# Latrophilin-2 mediates fluid shear stress mechanotransduction at endothelial junctions

Keiichiro Tanaka [ID] [1,7] [✉], Minghao Chen [ID] [1,7], Andrew Prendergast[1], Zhenwu Zhuang[1], Ali Nasiri[2], Divyesh Joshi [ID] [1], Jared Hintzen[1], Minhwan Chung [ID] [1], Abhishek Kumar [ID] [1], Arya Mani[1], Anthony Koleske [ID] [3], Jason Crawford[4], Stefania Nicoli[1] & Martin A Schwartz [ID] [1,5,6] [✉]

## Abstract

Endothelial cell responses to fluid shear stress from blood flow are crucial for vascular development, function, and disease. A complex of PECAM-1, VE-cadherin, VEGF receptors (VEGFRs), and Plexin D1 located at cell–cell junctions mediates many of these events. However, available evidence suggests that another mechanosensor upstream of PECAM-1 initiates signaling. Hypothesizing that GPCR and Gα proteins may serve this role, we performed siRNA screening of Gα subunits and found that Gαi2 and Gαq/11 are required for activation of the junctional complex. We then developed a new activation assay, which showed that these G proteins are activated by flow. We next mapped the Gα residues required for activation and developed an affinity purification method that used this information to identify latrophilin-2 (Lphn2/ADGRL2) as the upstream GPCR. Latrophilin-2 is required for all PECAM-1 downstream events tested. In both mice and zebrafish, latrophilin-2 is required for flow-dependent angiogenesis and artery remodeling. Furthermore, endothelial-specific knockout demonstrates that latrophilin plays a role in flow-dependent artery remodeling. Human genetic data reveal a correlation between the latrophilin-2-encoding *Adgrl2* gene and cardiovascular disease. Together, these results define a pathway that connects latrophilin-dependent G protein activation to subsequent endothelial signaling, vascular physiology, and disease.

**Keywords** Latrophilin; G Protein-coupled Receptor; Fluid Shear Stress; Vascular Development; PECAM-1
**Subject Categories** Signal Transduction; Vascular Biology & Angiogenesis

## Introduction

Fluid shear stress (FSS) from blood flow is critical for early vascular development in mice (Lucitti et al, 2007) and zebrafish (Sugden et al, 2017), for homeostatic vessel remodeling in adults (Humphrey and Schwartz, 2021; Vogel et al, 2020), and in the initiation and progression of atherosclerosis (Baeyens et al, 2016; Zhou et al, 2014). A complex of proteins at cell–cell junctions consisting of PECAM-1, VE-cadherin, VEGFR2 and 3, and Plexin D1 plays an important role in shear stress responses, including vasodilation, flow-dependent vessel remodeling, and atherosclerosis (Conway and Schwartz, 2012; Givens and Tzima, 2016). Stimulation of this junctional complex by flow triggers activation of Src family kinases (SFKs) within seconds, resulting in ligand-independent activation of VEGF receptors and downstream events, including endothelial cell (EC) alignment in the direction of flow (Collins et al, 2012). However, the mechanisms by which proteins located at cell–cell junctions can sense forces from shear stress exerted on the apical surface is unclear. Previous work suggests that activation of the junctional complex is not primary but rather requires an upstream event, mediated presumably by another mechanosensor (Conway et al, 2013).

GPCR/Gα protein signaling is an obvious candidate for initiating signaling on this time scale (seconds). GPCRs are a large family of cell surface receptors that play a crucial role in transducing signals across cell membranes. GPCRs undergo conformational changes in response to ligand binding or other stimuli, after which the activated intracellular domain binds to a heterotrimeric Gα protein complex and catalyzes the exchange of guanosine diphosphate (GDP) for guanosine triphosphate (GTP). Connections between GPCRs and shear stress have been reported (Chachisvilis et al, 2006; Dela Paz and Frangos, 2019; Erdogmus et al, 2019; Jung et al, 2012; Wang et al, 2015; Xu et al, 2018). However, the literature on Gα proteins and GPCRs in shear stress signaling is highly inconsistent (Chachisvilis et al, 2006; Dela Paz and Frangos, 2019; Jung et al, 2012; Wang et al, 2015). For instance, it has been demonstrated that flow applied to endothelial cells activates Gα proteins independent of an intervening GPCR (Dela Paz et al, 2017; Gudi et al, 1998). It has also been reported that shear stress responses related to PECAM-1 and cell–cell junction require GPCRs but with different GPCRs implicated in different studies. These studies include reports of ligand-independent activation of a bradykinin receptor (Chachisvilis et al, 2006; Yeh et al, 2008) and the sphingosine 1-phosphate receptor S1PR1 (Jung et al, 2012). The purinergic receptor P2Y2 was also implicated, in this case via ATP release downstream of activation of the mechanosensitive ion channel Piezo1

[1]Yale Cardiovascular Research Center, Section of Cardiovascular Medicine, Department of Internal Medicine, School of Medicine, Yale University, New Haven, CT 06511, USA. [2]Department of Internal Medicine, Yale University, New Haven, CT, USA. [3]Department of Molecular Biochemistry and Biophysics, Yale University, New Haven, CT, USA. [4]Department of Chemistry, Yale University, New Haven, CT, USA. [5]Department of Cell Biology, Yale University, New Haven, CT, USA. [6]Department of Biomedical Engineering, Yale University, New Haven, CT, USA. [7]These authors contributed equally: Keiichiro Tanaka, Minghao Chen. ✉E-mail: keiichiro.tanaka@yale.edu; martin.schwartz@yale.edu

(Albarran-Juarez et al, 2018; Wang et al, 2015), though this pathway has been disputed (Dela Paz and Frangos, 2019). The orphan receptor GPR68 was identified as a shear stress sensor specific to small arteries though the mechanism of activation, including ligand dependence or independence, is unknown (Xu et al, 2018). These studies also vary in the Gα proteins implicated, which included different Gαi isoforms (Gudi et al, 1996; Jung et al, 2012) and the Gαq/11 family (Albarran-Juarez et al, 2018; Chachisvilis et al, 2006; Wang et al, 2015).

To resolve this issue, we took an unbiased approach starting from the Gα proteins using the morphological changes in endothelial alignment to flow as the initial readout. Our results showed that Gαi2 and Gαq/11 function in parallel, identified latrophilin-2 (LPHN2) as the key upstream GPCR, and demonstrated a specific requirement for LPHN2 in flow-dependent remodeling in zebrafish and mice.

## Results

### Identification of Gα protein subunits required for flow sensing

To investigate the roles of GPCR signaling in endothelial flow responses, we first transfected human umbilical vein endothelial cells (HUVECs) with short interfering RNAs (siRNAs) to suppress Gs, Gi, Gq/11, and G12/13 classes of Gα proteins and assayed EC alignment in flow as our initial readout. Single knockdowns gave either weak (Gi) or no (Gs, G12/13, Gq/11) inhibition of EC alignment in flow (Fig. 1A; Appendix Fig. S1A–C). However, the simultaneous knockdown of Gi and Gq/11 completely inhibited alignment (Fig. 1A), whereas other combinations were ineffective (Appendix Fig. S1D,S1E). Activation of SFKs is the earliest flow-responsive event mediated by PECAM-1, followed by VEGFRs and Akt (Tzima et al, 2005). These events were similarly blocked by knockdown of Gi plus Gq/11 but not by single knockdowns (Fig. 1B,C; Appendix Fig. S1F). Therefore, Gi and Gq/11 act in parallel to initiate PECAM-1-dependent flow signaling.

Rescue controls with siRNA-resistant Gα variants showed that siRNA-resistant Gq and Gi2 restored cell alignment, whereas Gi1 and Gi3 did not (Fig. 1D; Appendix Fig. S2A,S2B). This unexpected result prompted us to search for residues that are similar in Gq and Gi2 but different in Gi1 and Gi3. Two residues (using human Gi2 numbering) fit this pattern: D167 in Gi2 corresponds to N166 in Gi1/3; and K307 corresponds to Q306 in Gi1/3 (Fig. 1E). Gi1(Q306K), but not Gi1(N166D), rescued EC alignment in flow (Fig. 1F; Appendix Fig. S2C). Consistently, the Gi2 mutant lacking K307 (K307Q) could not rescue the defects in flow-induced alignment after knockdown of Gi plus Gq/11 (Appendix Figs. S1G and S2D). Therefore, K307 in Gi2 distinguishes the Gα subunits that participate in flow signaling.

### Flow-induced activation of specific Gα proteins

K307 lies in the region that specifies coupling with GPCRs (Bae et al, 1999; Flock et al, 2017), suggesting that K307 may determine Gα activation. However, testing this hypothesis requires measurement of Gα activation in live cells. Unfortunately, indirect Gα activation assays using second messengers such as cAMP or calcium are unreliable in the context of fluid shear stress where

multiple activating/suppressing pathways are stimulated. We therefore developed pulldown assays for Gα based on the specific binding of GTP-loaded Gα subunits to effector proteins, namely GINIP for Gi and GRK2 N-terminal domain for Gq (Appendix Fig. S3a) (Gaillard et al, 2014; Tesmer et al, 2005). To facilitate these assays and distinguish specific Gα isoforms (especially Gi isoforms), we prepared versions containing an internal GluGlu (EE) epitope tag that does not impair function (Appendix Fig. S3B and refs (Medina et al, 1996; Wilson and Bourne, 1995)). For Gq, to enhance pulldown efficiency, we co-expressed Ric8A, which stabilizes active Gq (Papasergi et al, 2015). Active, but not inactive, Gi and Gq bound to these effector proteins, while when the artificial GPCR DREADD was expressed, its ligand clozapine-N-oxide (CNO) also activated Gα proteins (Appendix Fig. S3C–S3F). Importantly, this Gα pulldown assay showed not only rapid activation of Gi2 and Gq by FSS (Fig. 1G-J), but also differential responses of Gi1 and gain-of-function Gi1 (Q306K) to FSS (Fig. 1K,L). Gi activation was unaffected by PECAM-1 knockdown, indicating that this step was upstream or independent of PECAM-1 (Fig. 1M,N). Flow-mediated activation of specific Gα subunits thus corresponds to their requirement in EC flow responses.

### Identification of the upstream GPCR

We next sought to identify the GPCR(s) that mediate flow activation of Gi2 and Gq/11. Although affinity purification is the obvious choice, the rapid release of Gα proteins from GPCRs after GTP loading limits this approach. Starting with wild-type Gi1 and its Q306K mutant, we inserted (1) an internal GFP for affinity purification and (2) four alanines (hereafter ins4A) into helix α5, which blocks GTP loading, thereby preventing dissociation from GPCRs even in the presence of GTP (Kaya et al, 2016) (Fig. 2A). Both constructs localized mainly to the plasma membrane, as did wild-type Gα (Appendix Fig. S4A). We further employed detergent-free nanodisc-forming styrene-maleimide anhydride (SMA) copolymer to stabilize GPCR conformation within its native lipid environment (Lee et al, 2016). This approach was validated by co-immunoprecipitation using GFP-Trap® nanobody beads of Gα variants with DREADD GPCR, which show increased association after receptor activation (Appendix Fig. S4B,S4C). Proteomic analysis for GPCRs that associated with Gi1(Q306K) but not wild-type Gi1 in response to FSS (protocol in Appendix Fig. S4D), identified S1PR1 and ADGRL3 (Appendix Fig. S4E). However, depletion of S1PR1 did not block alignment in FSS (Appendix Fig. S4F,S4G). Furthermore, S1PR1 reportedly activates Gi1 and Gi3 as well as Gi2 (Lee et al, 1996), which is inconsistent with the Gα specificity defined above. We therefore focused on ADGRL family proteins, also called latrophilins (LPHNs).

Latrophilins (1, 2, and 3 in mammals) are adhesion-type GPCRs that bind to counterreceptors on neighboring cells and regulate neuronal synapses (Sando et al, 2019). Although it has been reported that different types of endothelial cells have different latrophilin isoforms, LPHN2 is the most widely expressed and is the overwhelming major isoform in HUVECs (Appendix Fig. S5A) (Maleszewska et al, 2016). Consistent with the proteomic results, Gi1(Q306K) (ins4A) associated with endogenous LPHN2, which increased upon FSS (Fig. 2B,C). This result strongly suggests that FSS activates LPHN2 to recruit Gα proteins. Furthermore, Gi1(Q306K), Gi2, and Gq co-immunoprecipitated with LPHN2,

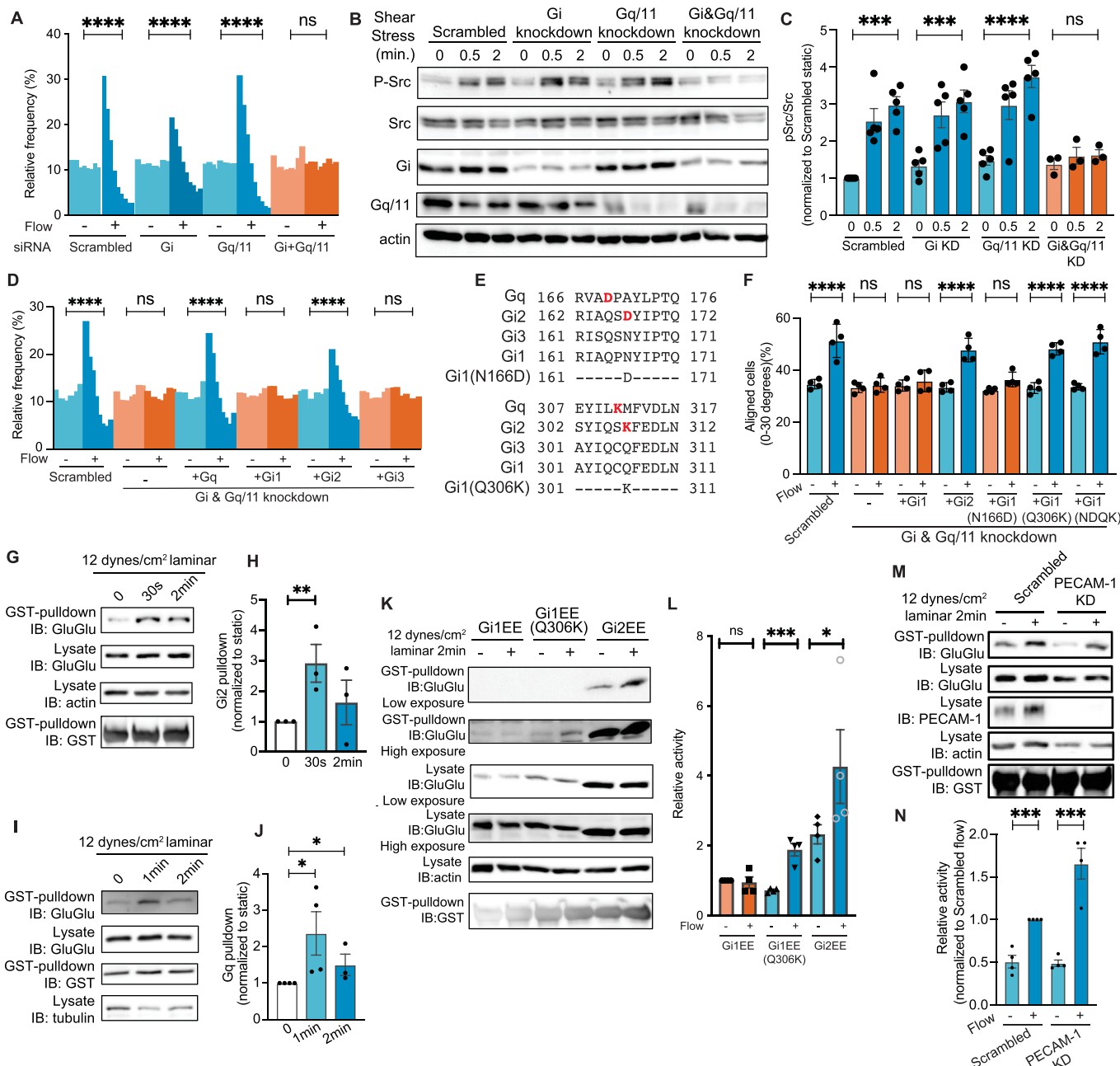

**Figure 1. Gα proteins specific for endothelial flow responses.**

(A) Flow-induced endothelial alignment after Gα knockdown. HUVECs were subjected to fluid shear stress (FSS) at a rate of 12 dynes/cm² for 16 h, and nuclear orientation quantified as histograms showing the percentage of cells within each 10° of the flow directions from 0° to 90° (see "Methods") ****P < 0.0001; one-way ANOVA with Tukey's multiple comparisons test. (B) Src family kinase activation, quantified in (C). n = 5 for control, Gi knockdown, Gq11 knockdown and n = 3 for simultaneous knockdown of Gi and Gq11. Values are means ± SEM. ****P < 0.0001, ***P < 0.001; one-way ANOVA with Tukey multiple comparison test. (D) Rescue of Gq/11 and Gi knockdown by re-expression of siRNA-resistant versions of the indicated proteins. ****P < 0.0001; one-way ANOVA with Tukey's multiple comparison test. (E) Amino acid sequences of Gαi1, Gαi2, and Gαi3 at the mutation sites of Gαi1 gain-of-function mutant. (F) Rescue of Gq/11 and Gi knockdown with indicated Gα proteins. Each point corresponds to one measurement averaged from >500 cells. N = 4. ****P < 0.0001; one-way ANOVA with Tukey's multiple comparisons test. (G) GINIP pulldown assay for activation of Gαi2 by FSS. N = 3. Results quantified in (H). **P = 0.0185, Student's t test. (I) GRK2N pulldown assay for activation of Gq. N = 4, quantified in (J). *P < 0.05, Student's t test. (K) GINIP pulldown assay for FSS-induced activation of wild-type and Q306K Gαi1. N = 4, quantified in (L). *P = 0.0304, ***P = 0.0017; Student's t test. (M) Gi2 activation after PECAM-1 knockdown. HUVECs expressing GluGlu-tagged Gi2 were transfected with scrambled siRNA or PECAM-1 siRNA, exposed to FSS, and Gi2 activation assayed as described above, quantified in (N). Values are means ± SEM, normalized to input Gα protein levels. ***P < 0.001; Student's t test, N = 4.

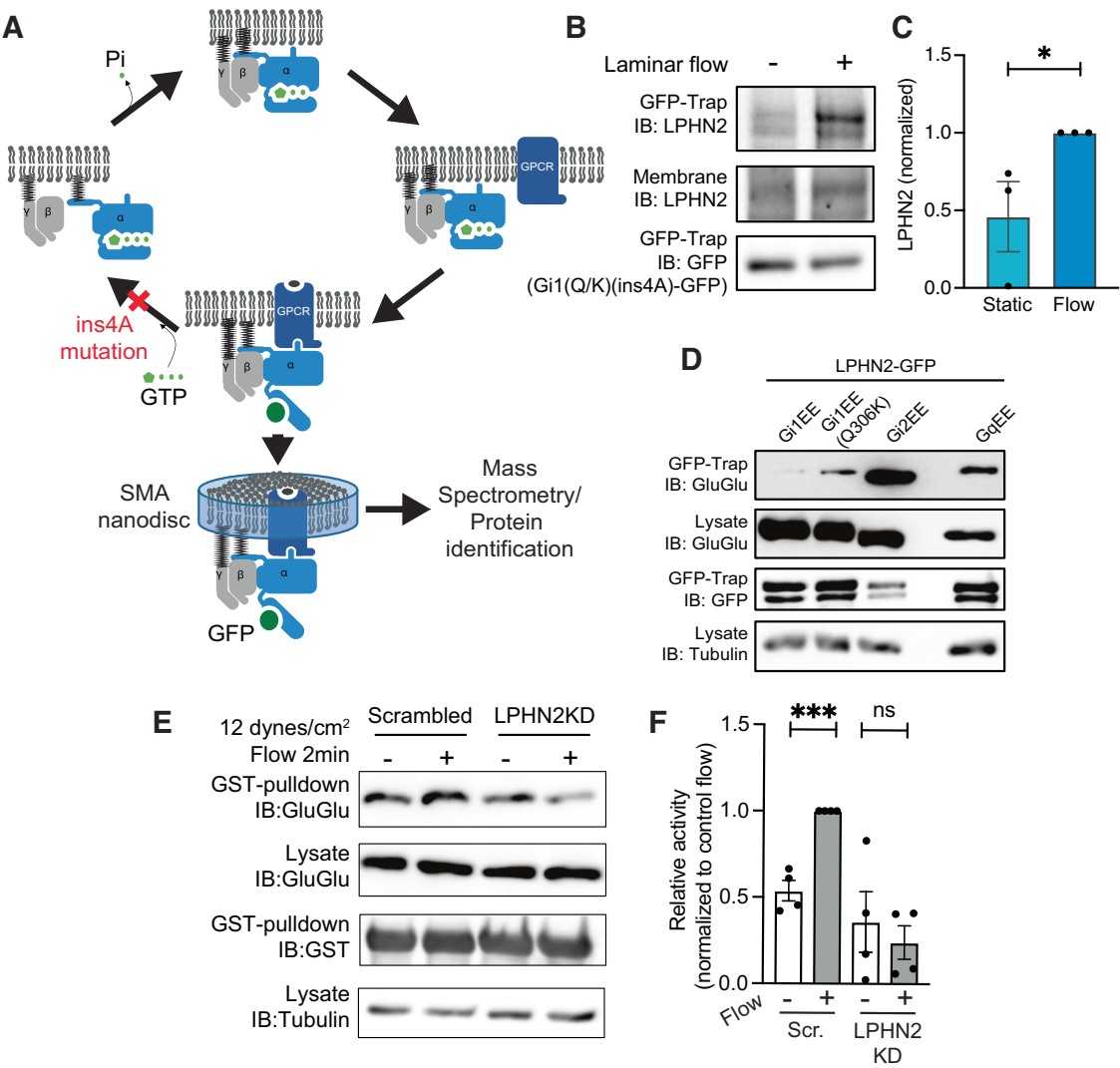

**Figure 2. Identification of the upstream flow-responsive GPCR.**

(A) Strategy for identification of GPCRs that bind the gain-of-function Gi1Q306K mutant but not the wild-type Gi1. (B) Co-immunoprecipitation of Gi1Q306K(ins4A) with endogenous LPHN2 with and without FSS for 2 min. $N = 3$, quantified in (C). *$P = 0.0374$, Student's $t$ test. (D) Co-immunoprecipitation of LPHN2-GFP with the indicated Gα proteins containing internal GluGlu epitope tags. (E) Gi2 pulldown assay after LPHN2 knockdown. $N = 4$. Results quantified in (F). ***$P < 0.001$, Student's $t$ test.

whereas Gi1 did not, indicating that LPHN2 specifically bound to the flow-responsive Gα proteins defined above (Fig. 2D). Importantly, LPHN2 knockdown blocked flow-mediated activation of Gi2 (Fig. 2E,F). Together, these data provide strong evidence that latrophilin-2 mediates Gα activation essential for endothelial alignment in response to flow.

## Latrophilins in EC flow signaling in vitro

Knockdown of LPHN2, but not LPHN1 or LPHN3, blocked EC alignment in flow (Fig. 3A,B; knockdown confirmed in Appendix Fig. S5B,S5C). LPHN2 knockdown in human aortic ECs similarly blocked flow-induced alignment (Appendix Fig. S5D,S5E). In line with these results, LPHN2 knockdown also blocked Golgi apparatus polarization in flow (Appendix Fig. S5F,S5G). LPHN2 localized to cell–cell contacts (Fig. 3C; Appendix Fig. S6). The flow-responsive

G1 mutant, Gi1(Q306K), also associated with PECAM-1 following FSS (Fig. 3D). The association between the LPHN2-Gi1(Q306K) complex and the junctional mechanosensory complex proteins PECAM-1 and VE-Cadherin was detected after 24 h of flow (Fig. 3D–G), indicating a persistent interaction. We also assessed the immediate flow-dependent signals. LPHN2 depletion strongly suppressed flow-mediated acute activation of SFKs and Akt (Fig. 3H–J). Thus, LPHN2 colocalizes, physically associates with and is required for signaling through the junctional complex. However, LPHN2 knockdown did not inhibit activation of VEGFR2 by VEGF-A, indicating that LPHN2 functions specifically in response to laminar flow (Appendix Fig. S7A,S7B).

Among the three latrophilin isoforms in the human genome, all rescued flow-mediated endothelial alignment after LPHN2 knockdown (Fig. 3K; expression confirmed in Appendix Fig. S8A), indicating functional equivalence; the observed differences are

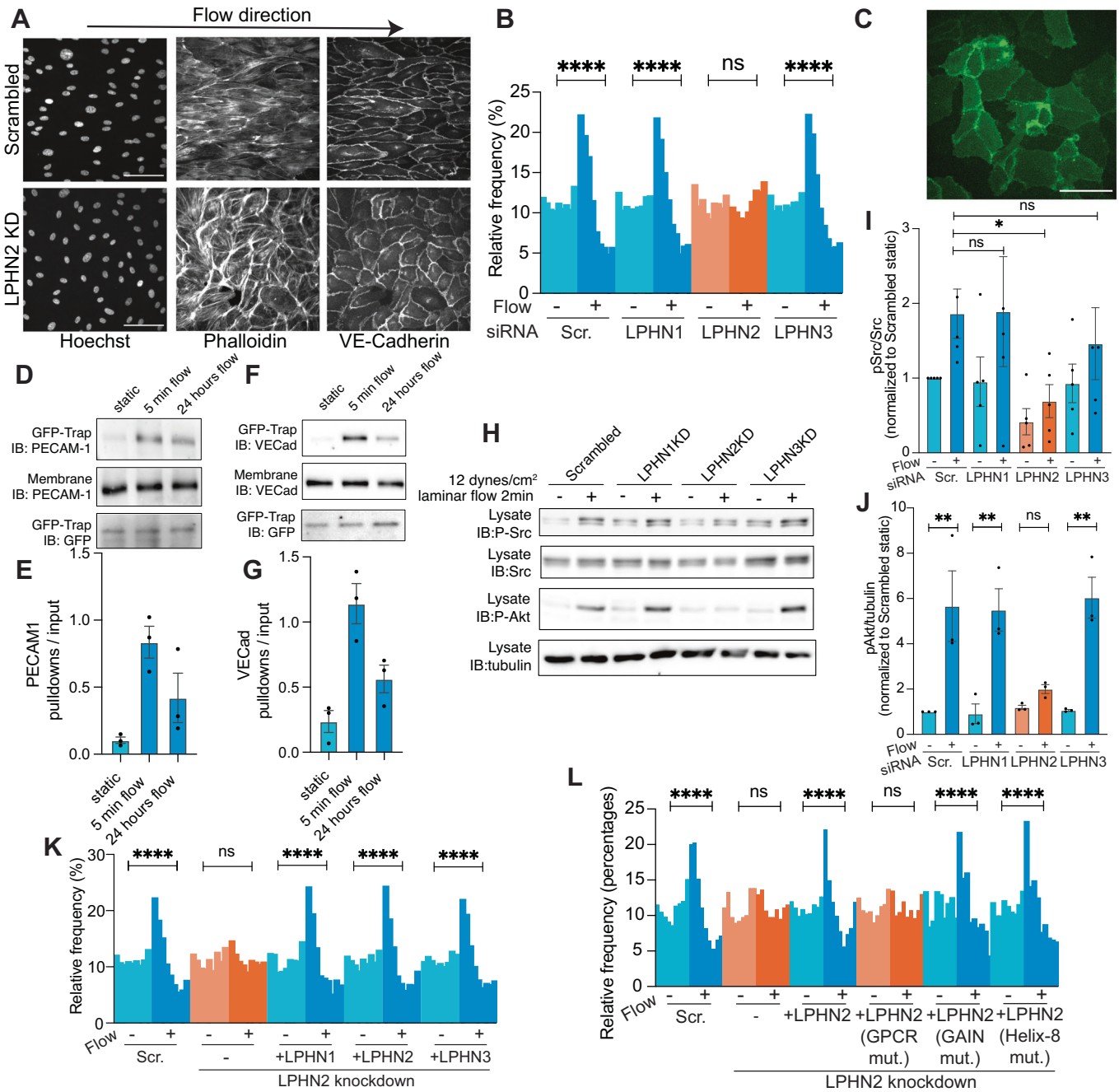

**Figure 3. Latrophilins regulate endothelial flow responses.**

(A) HUVECs with or without latrophilin-2 knockdown were subjected to FSS for 16 h, then fixed and stained with Hoechst (nuclei), phalloidin (F-actin) and an antibody against VE-Cadherin. Scale bar: 100 μm. (B) Alignment of HUVECs (each bar = 10° increments) after knockdown of latrophilin isoforms was quantified as in Fig. 1 from >2000 cells/experiment, N = 3. ****P < 0.0001; one-way ANOVA with Tukey's multiple comparisons test. (C) Localization of LPHN2-mClover3. Scale bar: 100 μm. N = 6. (D) Gi1(Q306K) pulldown from ECs ± FSS for 5 min or for 24 h, probed for PECAM-1. N = 3, quantified in (E). (F) Gi1(Q306K) pulldown from ECs ± FSS for 5 min or for 24 h probed for VE-cadherin. N = 3, quantified in (G). (H) Activation of Src family kinases and Akt by FSS in HUVECs depleted for latrophilin isoforms. N = 3–4, quantified in (I, J). *P = 0.0184, **P < 0.001; one-way ANOVA with Tukey's multiple comparisons test. (K) Alignment of LPHN2-depleted HUVECs rescued by re-expression of the indicated latrophilin isoforms. ****P < 0.0001; one-way ANOVA with Tukey's multiple comparisons test. Quantification of >2000 cells for each condition, N = 3. (L) LPHN2 knockdown HUVECs were rescued by re-expression of the indicated LPHN2 mutants. ****P < 0.0001, one-way ANOVA with Tukey's multiple comparisons test. Quantification of >500 cells for each condition.

therefore due to differential expression. EC alignment was rescued by wild-type (WT) LPHN2 but not H1071A, which is defective in coupling to Gα proteins (Nazarko et al, 2018) (Fig. 3L; expression confirmed in Appendix Fig. S8B). Adhesion-type GPCRs can be activated by internal cleavage by the extracellular GAIN domain which exposures its *Stachel* peptide as a tethered agonist (Vizurraga et al, 2020). However, the autoproteolysis-defective mutant fully rescued EC alignment (see Fig. 3L). Helix-8 is known to endow some GPCRs with mechanosensitivity to membrane stretch (Erdogmus et al, 2019). However, a LPHN2 helix-8 mutant could still rescue EC alignment in flow (Fig. 3L). Taken together, these findings indicate that latrophilin regulates Gα activation required for endothelial flow responses, but activation requires neither the *Stachel* peptide nor helix-8.

## Latrophilin-2 regulates flow-induced endothelial cell morphological changes in vivo

Long-term laminar flow stabilizes EC junctions, promoting a linear, continuous morphology associated with barrier function (Kroon et al, 2017) (see Fig. 3A). To test the role of LPHN2 in this process, we analyzed junctional morphology in vitro with and without 16 h of flow. In control cells, shear stress shifted junctions from an irregular morphology to linear junctions; in Lphn2 knockdown ECs, junctions were indistinguishable in the absence of flow but failed to linearize in flow (Fig. 4A,B), demonstrating that the effect is flow-specific. To further examine contributions of blood flow/latrophilin-2 to ECs in vivo, zebrafish embryos are useful as a developmental system that is amenable to manipulation of shear

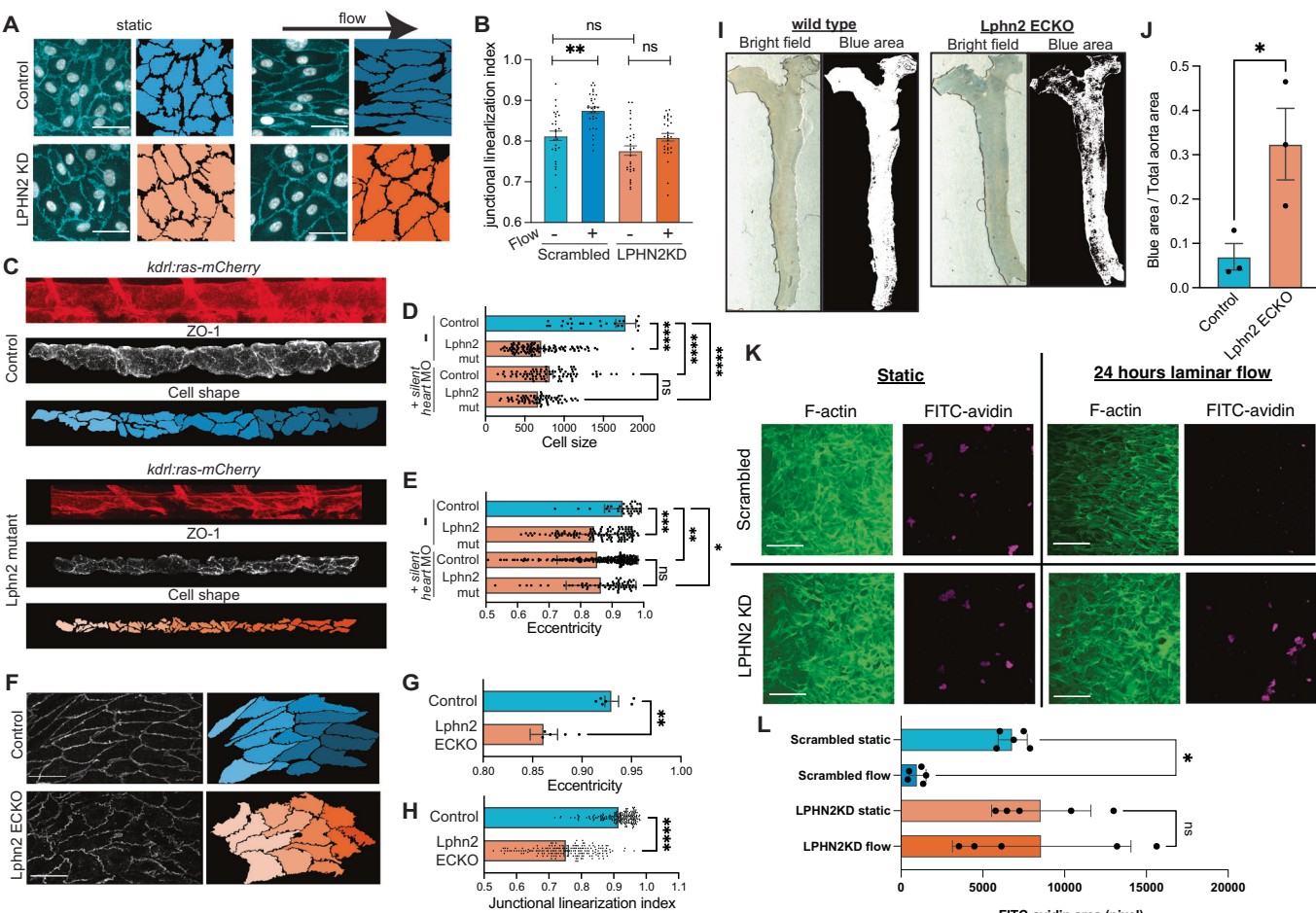

**Figure 4.  Latrophilins in flow-mediated endothelial cell morphology in vivo.**

(A) Representative image of HUVECs transfected with scrambled siRNA or LPHN2 siRNA. Outlines of ECs from VE-cadherin staining are shown to next to each image. Scale bar: 100 μm. (B) Quantification of junctional linearization index from (A). $n = 30$ for each condition. **$P = 0.0019$, Statistics used two-way ANOVA with Sidak's multiple comparison test. (C) Dorsal aortas from 48 hpf Tg(*kdrl:ras-mCherry*) WT or LPHN mutant zebrafish embryos stained for ZO-1. EC were outlined as in Methods. LPHN2 mutant indicates *adgrl2a+adgrl2b.1* sgRNAs. (D, E) Quantification of cell size and eccentricity in embryos with latrophilin-2 CRISPR sgRNAs ± *silent heart* morpholinos (MO). $N = 6$ per condition. ****$P < 0.0001$, *$P = 0.0267$, **$P = 0.0015$, ***$P = 0.0008$; one-way ANOVA with Tukey's multiple comparisons test. (F) ECs in the thoracic aorta in 18-week-old mice at 10 weeks after tamoxifen injections, stained for β-catenin (left) and outlined (right). Data are representative of 6 mice for each condition. Scale bar: 100 μm. (G, H) Quantification of cell eccentricity and junction linearity in mouse aorta. $N = 4$ per condition. **$P = 0.0013$; ****$P < 0.0001$; Student's $t$ test. (I) Aortas from mice 30 min after Evans blue dye injection, thresholded images on the right. The fractional blue area quantified in (J). $n = 3$ for each condition. *$P = 0.0419$; Student's $t$ test. (K) In vitro endothelial permeability assay using FITC–streptavidin and biotin-conjugated fibronectin. Scale bar: 100 μm. (L) Permeability was quantified by measuring FITC–streptavidin area per image field from (K). $n = 5$, *$P = 0.0457$, one-way ANOVA.

stress using morpholinos against cardiac troponin T type 2a (tnnt2a; "*silent heart*") (Sehnert et al, 2002), without global adverse effects. We employed zebrafish embryos expressing Tg(*kdrl:ras-mCherry*) to mark endothelial cells (Sehnert et al, 2002) and designed sgRNAs for CRISPR-Cas9 targeting all latrophilin isoforms (*adgrl1*, *adgrl2a* plus *adgrl2b.1*, and *adgrl3*). Our sgRNAs induced >50% cleavage in the T7 endonuclease assays and reduced mRNA levels through nonsense-mediated decay (Appendix Fig. S9). Since Lphn2 is the major endothelial isoform, we performed CRISPR-mediated F(0)/mosaic knockout of *adgrl2a+adgrl2b.1* (hereafter Lphn2 mutation), which inhibited elongation and enlargement of ECs in the dorsal aorta (Fig. 4C). Injecting single-cell embryos with *silent heart* morpholinos at a dose that completely blocked cardiac contractility similarly blocked the elongation of ECs. Furthermore, combining tnnt2a and Lphn2 mutation had no further effect, supporting a specific role for latrophilins in flow sensing (Fig. 4D,E).

To address LPHN function in mammals, we crossed Lphn2$^{flox/flox}$ mice with CDH5-Cre$^{ERT2}$ mice (Lphn2 ECKO), which were injected with tamoxifen at 8 weeks of age and remained viable 10 weeks later. The descending thoracic aorta (a region of high laminar flow) of Lphn2 ECKO mice had poorly aligned ECs with irregular junctions (Fig. 4F–H). In addition, Lphn2 ECKO increased vascular permeability in the whole aorta, as assessed by Evans blue extravasation (Fig. 4I,J). In vitro, flow for 24 h increased HUVEC barrier function, which was blocked by LPHN2 knockdown, whereas barrier function in the absence of flow was unaffected (Fig. 4K,L). Latrophilins are therefore flow-responsive GPCRs that regulate multiple steps in PECAM-1-dependent EC mechanotransduction.

## Plasma membrane fluidization initiates latrophilin signaling

We next considered how FSS activates LPHN2. As mutation of the autoproteolytic GAIN domain, which is speculated to be mechanosensitive, had no effect (Fig. 3I), we considered alternatives. FSS induces a rapid increase in plasma membrane fluidity, in some cases associated with decreased plasma membrane cholesterol (Butler et al, 2001; Haidekker et al, 2000; Yamamoto and Ando, 2013; Yamamoto et al, 2020). As GPCRs can respond to such changes (Vizurraga et al, 2020), HUVECs were treated with methyl-β-cyclodextrin (MβCD) to extract plasma membrane cholesterol. Treatment with 5 mM MβCD for 1 min reduced plasma membrane cholesterol to the same extent as flow, assayed by binding of the cholesterol-specific probe D4H-Clover3 (Liu et al, 2017) (Fig. 5A). No discernable effect on cytoskeletal organization was observed under these conditions (Fig. 5B). MβCD treatment also induced Gα binding to LPHN2 (Fig. 5C) and activated both SFKs and VEGFR2, which were blocked by LPHN2 knockdown (Fig. 5D–F). Together with published data demonstrating that cholesterol supplementation blocks flow signaling to these and related pathways (Fancher et al, 2018; Tirziu et al, 2005; Yamamoto and Ando, 2013, 2015; Yamamoto et al, 2020), these results suggest that membrane fluidization can trigger LPHN2 GPCR activation and downstream signaling (Fig. 5G).

## Latrophilins in vascular remodeling in vivo

To address the roles of LPHN2 in flow-dependent vascular remodeling, we examined zebrafish embryos, where artery lumen diameters are determined by EC shear stress (Sugden et al, 2017). We observed that intersegmental vessels (ISVs) at 48 h post fertilization had reduced diameters after both latrophilin knockout and blockade of blood flow, with no further effect when combined (Fig. 6A,B). At 72 hpf, when intersegmental arteries and veins are distinguishable, the decrease in diameter was observed only in arteries (Fig. 6C). LPHNs are thus required for the effects of flow on artery diameter.

These results prompted us to examine flow-dependent vascular remodeling in mice. Femoral artery ligation triggers flow-driven arteriogenesis in the thigh and hypoxia-driven angiogenesis in the calf (Cornella et al, 2017). Following surgery, LPHN2 ECKO mice showed lower blood flow in the foot even at day 1 and markedly slower recovery than WT littermates (Fig. 6D,E). To understand these effects, we performed micro-CT of vascular casts after injection of electron-dense bismuth microparticles that label arteries/arterioles but not capillaries due to particle size. Surgery induced an increase in small arteries in WT mice but essentially none in LPHN2 ECKO mice (Fig. 6F,G). LPHN2 ECKO mice also had a much lower density of small arteries even in the control leg, consistent with lower flow on day 1. Staining for PECAM to examine the total vasculature in leg muscle sections revealed decreased capillary density as well, even at baseline (Fig. 6H,I). LPHN2 ECKO mice thus show low vascular density at baseline and greatly reduced flow-dependent remodeling. As expected from the low vascular density, in a treadmill fatigue test (Fig. 6J) LPHN2 ECKO mice showed markedly lower exercise capacity (Fig. 6K,L) and impaired oxygen consumption (Fig. 6M). LPHN2 ECKO thus results in vascular defects in vivo, at least in part due to defective EC flow sensing. To address whether these defects were related to shear stress sensing, we examined an in vitro system in which FSS enhances VEGF-induced sprouting into collagen gels (Galie et al, 2014). Enhanced sprouting was completely blocked by LPHN2 depletion with only a weak effect on basal migration (Fig. 6N–P).

## Genetic link to human disease

The Cardiovascular Disease Knowledge Portal reported four SNPs near the 5' end of the *ADGRL2* gene (Appendix Fig. S10A) that exhibit genome-wide association with hypertension (Appendix Fig. S10B). All four are in close proximity to each other and are in linkage disequilibrium (D': 0.9953, R2: 0.97). Two of the SNPs meet the significant threshold ($P < 5 \times 10^{-8}$), and rs1195871 is relatively conserved among vertebrates. Of particular interest, the intronic SNP rs186892211 in the *ADGRL2* locus (Appendix Fig. S10A) is associated with large-artery atherosclerosis and subsequent ischemic stroke (TOAST classification) with a very high odds ratio (odds ratio 720; $P < 3.134e\text{-}10$, Appendix Fig. S10B).

# Discussion

These data elucidate a pathway in which fluid shear stress acts through the latrophilin adhesion GPCRs to activate subsequent cellular signaling events and vascular remodeling in vivo, thus connecting time scales from seconds to hours to weeks/months. Previous studies showed that laminar flow induces an increase in plasma membrane fluidity associated with depletion of membrane cholesterol (Yamamoto and Ando, 2013, 2015; Yamamoto et al, 2020).

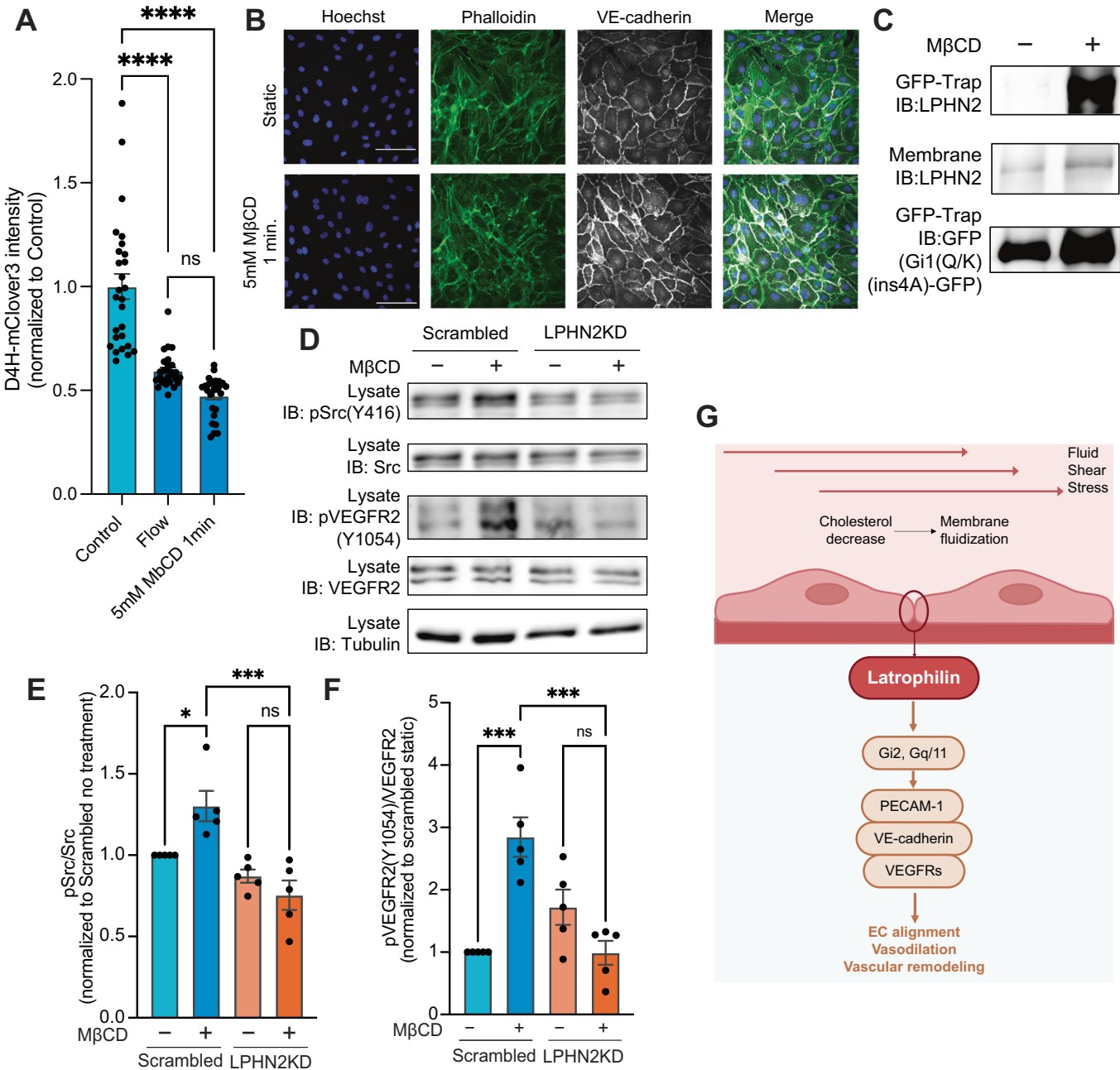

**Figure 5. Cholesterol depletion activates Latrophilin-dependent signaling.**

(A) ECs were treated with FSS or 5 mM methyl-β-cyclodextrin (mβCD) for 1 min then incubated with D4H-mClover3, rinsed and bound mClover3 quantified as described in "Methods". N = 27 for each condition. ****P < 0.0001; one-way ANOVA with Tukey multiple comparisons test. (B) Phalloidin staining following short acute treatment of methyl-β-cyclodextrin (mβCD). Scale bar: 100 μm. (C) Co-immunoprecipitation of Gi1Q306K(ins4A) with endogenous Latrophilin-2 (LPHN2) after 5 mM MβCD for 1 min. (D) Activation of Src family kinases and VEGFR2 after 5 mM MβCD for 1 min in HUVECs ± LPHN2 knockdown. Results quantified in (E, F), respectively. Values are means ± SEM. N = 5. *P = 0.0305, ***P = 0.0002; one-way ANOVA with Tukey multiple comparison test. (G) Model for junctional endothelial shear stress mechanotransduction.

Our data demonstrate that artificial cholesterol depletion activates flow-induced signals in an Lphn2-dependent manner, suggesting that latrophilins connect physical changes in the plasma membrane to cellular signaling. Latrophilins mediate activation of the PECAM-1/VE-cadherin/VEGFR pathway through both Gi2 and Gq/11, suggesting parallel or redundant downstream pathways.

Consistent with these functional effects, FSS induces physical interactions (direct or indirect) between LPHN2-Gα and PECAM-1. However, LPHNs are not required for activation of VEGFRs by its ligand VEGF.

Previous studies identified either Gi or Gq/11 as mediators of flow signaling (Chachisvilis et al, 2006; Jung et al, 2012; Wang et al, 2015).

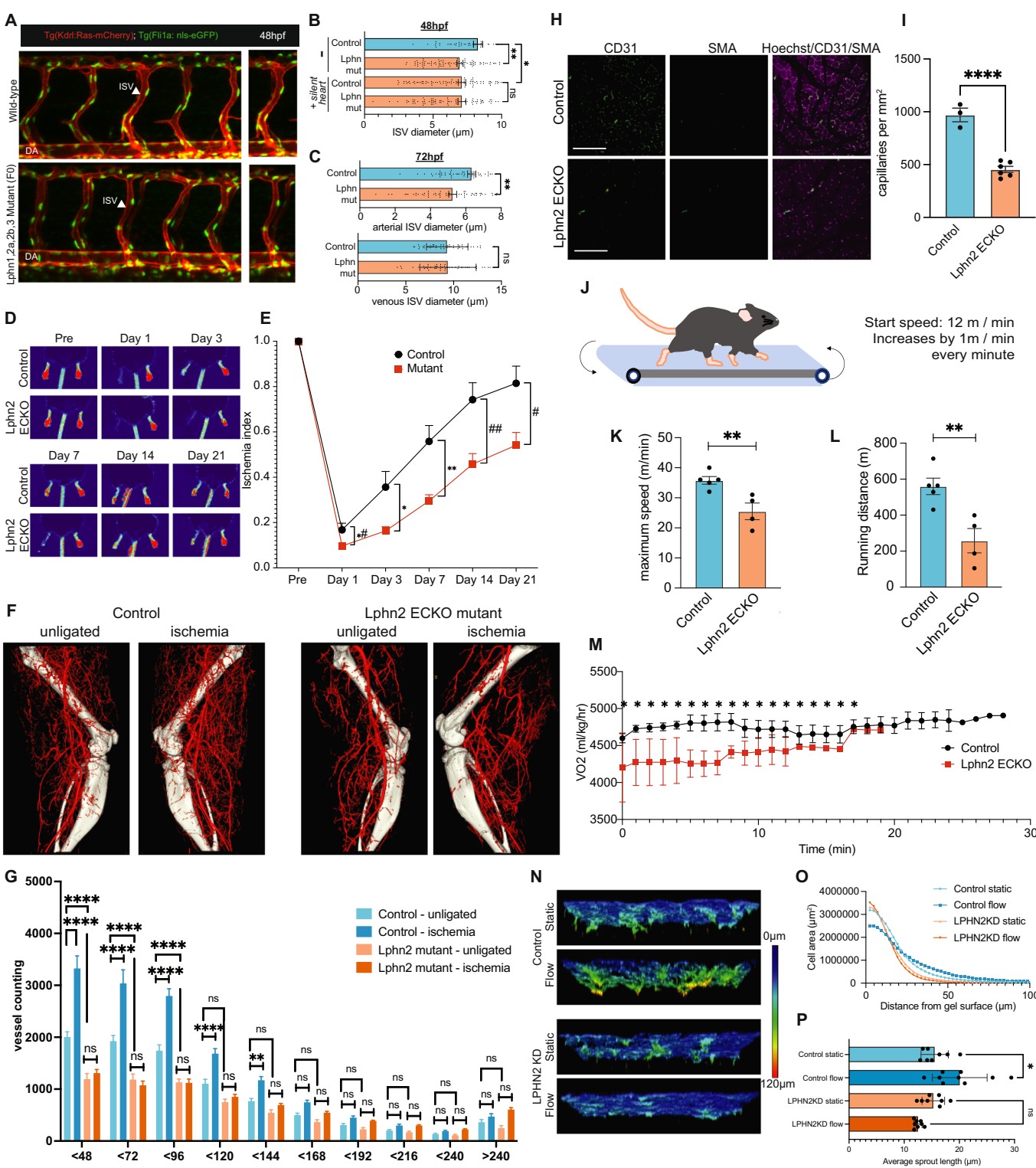

It seems plausible that, depending on EC subtype or experimental conditions, either Gi or Gq/11 predominated, which concealed their functional redundancy. Previous studies also identified other GPCRs as mediators of EC shear stress responses, via direct or indirect mechanisms, as well as making claims for GPCR-independent activation of G proteins (Dela Paz et al, 2017). GPCRs implicated in shear stress signaling include bradykinin receptor B2 (Chachisvilis et al, 2006), ATP receptor P2Y2 (Albarran-Juarez et al, 2018; Wang et al, 2015), sphingosine phosphate receptor S1PR1 (Jung et al, 2012), the apelin receptor

**Figure 6. Latrophilin-2 in flow-induced vascular remodeling.**

(A) Representative images of ISVs from 48 hpf embryos with F(0)/mosaic depletions of latrophilin genes or controls. (B) Quantification of diameters of all ISVs in (A). $n = 15$ per condition. *$P = 0.0452$, **$P = 0.0150$; one-way ANOVA with Tukey's multiple comparisons test. (C) Quantification of diameters of arterial and venous ISVs at 72 hpf of zebrafish. $n = 6$ per condition. **$P = 0.0013$; Student's $t$ test. (D) Ratio of blood flow in the right ischemic vs left control foot by laser doppler, quantified in (E). $N = 7–8$ per condition. *,#$P = 0.0439$, *$P = 0.0135$, **$P = 0.0028$, ##$P = 0.0056$, #$P = 0.0163$; Student's $t$ test. (F) Micro-CT of arterial vasculature after surgery. Representative reconstructed micro-CT images from LPHN2 ECKO and WT littermates on day 22 after surgery, quantified in (G). $n = 8$ per condition. ****$P < 0.0001$, ***$P = 0.0007$, *$P = 0.0167$; two-way ANOVA with Tukey's multiple comparisons test. (H) Rectus femoris muscle from thighs of 10-week-old mice of LPHN2 ECKO or wild-type littermates, stained for CD31 and SMA to visualize the entire vasculature and arteries. Scale bar: 100 μm. (I) Quantification of capillary density in the entire stitched thigh muscle images. $n = 3$ for control and $n = 6$ for Lphn2 ECKO. ****$P < 0.0001$; Student's $t$ test. (J) Schematic of mouse treadmill fatigue test. (K) Maximum running during test. $n = 4$ for control and $n = 5$ for Lphn2 ECKO. **$P = 0.0082$; Student's $t$ test. (L) Running distance for each mouse. $n = 4$ for control and $n = 5$ for Lphn2 ECKO. **$P = 0.0064$; Student's $t$ test. (M) Oxygen consumption during treadmill fatigue test. *$P < 0.0001$; two-way ANOVA with Tukey's multiple comparisons test. Control: $N = 5$, LPHN2 ECKO: $N = 4$. (N) Representative 3D reconstructed images at 20° downward angle of ECs plated on top of collagen gels and sprouting under static or flow conditions. ECs were stained with phalloidin; pseudocolors indicate distance from the gel surface as per the color scale on the right. (O) Histograms of cell area vs. distance from gel surface in (N). (P) Mean sprout length. $N = 4$. ***$P = 0.0006$; one-way ANOVA with Tukey's multi-comparison test.

APJ (Strohbach et al, 2018) and GPR68 (Xu et al, 2018). An indirect association of Gq/11 with PECAM-1 was also reported, which dissociated upon application of shear stress (dela Paz et al, 2014). Differences in EC location or subtype likely account for some of these effects. For example, GPR68 is expressed specifically in small resistance arteries where it controls vasodilation but had no reported effect on development or patterning (Xu et al, 2018). P2Y2 was also reported to affect vasodilation and blood pressure without reported effects on development or patterning. It seems likely that differences in local expression may explain some of the apparent multiplicity, though additional variables such as modulation by other inputs and coupling to different downstream effectors may also contribute. Understanding this GPCR-Gα protein signaling network is therefore an important though complex direction for future work. The novel pulldown assay and GPCR-Gα protein affinity purification protocol developed here may facilitate such efforts.

It is also noteworthy that latrophilins are required only for activation of signaling events, including VEGFR2 by flow, but not for activation of VEGFR2 by its ligand, VEGF. PECAM-1 first appears in vertebrates but has multiple other functions in leukocyte transmigration and activation, platelet reactivity and angiogenesis that are independent of flow (Privratsky et al, 2010). Together, these findings argue that latrophilins, in essence, confer flow sensitivity to pre-existing pathways controlled by ligand availability.

Lphn2 mutation in zebrafish and endothelial knockout in mice caused multiple vascular defects that implicate FSS mechanotransduction, including reduced EC alignment in the direction of flow, altered junctional morphology and permeability, smaller artery diameters and defects in arterialization in the hindlimb ischemia model. Lower capillary and small artery density in some vascular beds may also be attributable to defective FSS sensing, as FSS regulates angiogenic sprouting (Galie et al, 2014) and stabilizes nascent vessels once blood begins to flow (Campinho et al, 2020). We note, however, that flow-independent functional roles for Lphn2 in ECs have been observed (Camillo et al, 2021) and likely contribute as well.

These results link together many observations to provide a comprehensive model for EC responses to FSS (Appendix Fig. S11). These findings are of further interest in light of human genetic data that strongly link the *Adgrl2* gene to vascular disease (Appendix Fig. S10). However, the identified polymorphisms reside in

regulatory or intronic regions, thus, causal connections to vascular disease are not obvious. Intronic polymorphisms can regulate gene expression but could also be in linkage disequilibrium with causal SNPs that are not yet identified. More detailed population genetic analyses are required to address these issues.

These findings open many important questions for future work. Structural insights into latrophilin activation are needed, as are insights into the contribution of the large extracellular domain. The parallel roles of Gi2 and Gq/11 and downstream effectors remain to be understood. A thorough analysis of latrophilin functions in vascular development and physiology remains to be done. Lastly, combined human genetic and structure–function studies are needed to elucidate the contribution of Lphn2 variants to human cardiovascular disease.

# Methods

## Antibodies

Primary antibodies used in this study:

| Epitope | Vendor | Catalog # |
|---|---|---|
| Phospho-Src | Cell Signaling | 6943 |
| Src | Cell Signaling | 2109 |
| Gi | NewEast Biosciences | 26003 |
| Gq | NewEast Biosciences | 26060 |
| | BD Bioscience | 612705 |
| Gs | NewEast Biosciences | 26006 |
| Phospho-VEGFR2(Tyr1175) | Cell Signaling | 2478 |
| Phospho-VEGFR3 (Tyr1054/Tyr1059) | Invitrogen | 44-1047G |
| VEGFR2 | Cell Signaling | 2479 |
| Phospho-Akt (Ser473) | Cell Signaling | 9271 |
| VE-Cadherin | Santa Cruz | sc-6458 |
| LPHN2 | Invitrogen | PA5-65359 |
| RFP | Antibodies-Online | ABIN129578 |
| Alpha-smooth muscle actin | Invitrogen | 50-9760-82 |

| Epitope | Vendor | Catalog # |
|---------|--------|-----------|
| Alpha-smooth muscle actin-Cy3-conjugated | Sigma | C6198 |
| ZO-1 | Invitrogen | 61-7300 |
| Beta-catenin | Cell Signaling | 9562 |
| GluGlu | BioLegend | MMS-115P |
| HA | BioLegend | 901501 |
| GFP | Abcam | ab13970 |
| | Santa Cruz | sc-9996 |
| GST | Cell Signaling | 2625 |
| Tubulin | Invitrogen | 62204 |
| Actin | Santa Cruz | sc-8432 |
| PECAM-1 | Kind gift from Dr. Peter Newman | |
| | BD Bioscience | 557355 |
| GM130 | BD Bioscience | 610823 |

## Cell culture

Primary HUVECs were obtained from the Yale Vascular Biology and Therapeutics core facility. Each batch is composed of cells pooled from three donors. Cells were cultured in M199 (Gibco: 11150-059) supplemented with 20% FBS, 1× penicillin–streptomycin (Gibco: 15140-122), 60 µg/ml heparin (Sigma: H3393), and endothelial growth cell supplement (hereafter, complete medium). HUVECs used for experiments were between passages 3 and 6.

## Shear stress

For generating protein lysates for pulldown assays, HUVECs were seeded on tissue culture-treated plastic slides coated with 10 µg/ml fibronectin for 1 h at 37 °C and then grown to confluence in HUVEC complete media. For short-term stimulation with shear stress, cells were starved overnight in M199 medium with 2% FBS and 1:10 of ECGS or for 30 min in M199 medium containing 0.2% BSA. These slides were set in parallel flow chambers, and shear stress was applied as described (Frangos et al, 1988).

For imaging cells under shear stress, HUVECs were plated in HUVEC complete medium on glass-bottom six-well plates coated with 10 µg/ml fibronectin and shear stress applied on an orbital shaker at 150 rpm for 24 h, leading to uniaxial pulsatile shear in the peripheral region and multiaxial shear in the middle of the well, as previously described (Arshad et al, 2021). Cells in the outer region, 0.7–0.9 cm from the center of each well were examined for Golgi orientation and endothelial alignment.

## Image analysis

Cell orientation was calculated by taking the masks of the cell nuclei determined by Hoechst images, fitting them as an ellipse, and determining the angle between the flow direction and the major axis of the ellipse. Analyzed results were visualized as histograms showing the percent of cells within each 10° of the direction of flow or as quantification of aligned cells with nuclei whose major axis were within 0–30° to flow direction as indicated in the figure legends.

Eccentricity for cell alignment was calculated based on first eccentricity of the ellipse following the equation below.

$$Eccentricity = \frac{\sqrt{a^2 - b^2}}{a}$$

(a: major axis, b: minor axis)

The junctional linearization index was calculated as ratio of perimeter of the cell to perimeter of its convex hull following the equation below:

$$Junctional\ linearization\ index = \frac{exact\ contour\ of\ the\ cell}{convex\ perimeter}$$

## Data display

Quantified data are displayed as means ± standard error (SEM) in which the indicated replicates are independent experiments. $F$ test was conducted to check equal variance.

## Lentiviral transduction

Lenti-X 293T cells (Clontech, 632180) were cultured for at least 24 h in DMEM supplemented with 10% FBS and lacking antibiotics, then transfected with lentiviral plasmids encoding the gene of interest and packaging plasmids (Addgene: 12259 and 12260) using Lipofectamine 2000 (Thermo Fisher Scientific: 11668-019) following the manufacturer's protocols with Opti-MEM medium. Conditioned media from these cultures were collected 48 h later, sterilized through 0.22-µm filters, and added to HUVECs together with 8 µg/ml of polybrene (Sigma: 107689). After 24 h, cells were switched to complete medium for 48 h.

## siRNA transfection

HUVECs were cultured in EGM™-2 Endothelial Cell Growth Medium-2 BulletKit™ (Lonza: CC-3156 and CC-4176) for 24 h before transfecting with RNAiMax (Thermo Fisher Scientific: 13778-150) with 20 nM siRNA in Opti-MEM (Gibco: 31985-070) according to the manufacturer's instructions. After 6 h, cells were switched to EGM-2 medium and used for experiments 2–3 days later. Gα protein siRNAs were custom-designed based on previous publications (Grzelinski et al, 2010; Krumins and Gilman, 2006; Ngai et al, 2008). For latrophilin knockdown, ON-TARGET plus Smartpool siRNAs from Dharmacon against human LPHN1 (L-005650-00-0005), LPHN2 (L-005651-00-0005), LPHN3 (L-005652-00-0005) were used.

## Western blotting

HUVECs were washed with PBS and extracted in Laemmli sample buffer. Samples were separated by SDS-PAGE and transferred onto nitrocellulose membranes. Membranes were blocked with 5% milk in TBS-T and probed with primary antibodies at 4 °C for overnight. The targeting proteins were visualized by HRP-conjugated secondary antibodies and subsequent HRP-luminol reaction.

## Immunofluorescence

HUVECs were washed with PBS, fixed for 10 min with 3.7% PFA in PBS. Following fixation, cells were permeabilized with 0.5% Triton X-100 in PBS for 10 min and then incubated with 3% BSA in PBS for 30 min for blocking. Cells were washed with PBS after blocking and were incubated Alexa488-Phalloidin and Hoechst for 1 h, then washed 4 times with PBS and mounted. Images were captured with ×20 or ×60 objective on a PerkinElmer spinning disk confocal microscope or a Leica Sp8 confocal microscope with the Leica Application Suite (LAS) software. Cell alignment was determined as described previously (Baeyens et al, 2014).

## Reverse transcription and quantitative PCR

RNA was isolated from HUVECs using RNeasy kit according to the manufacturer's instructions and quantified using a nanodrop spectrophotometer. Following cDNA synthesis using Bio-Rad iScript kit, RT-PCR was performed as follows. Each PCR reaction contains 42 two-step amplification cycles consisting of: (a) denaturation at 95 °C for 30 s, and (b) annealing and extension at 60 °C for 30 s. The amplification curve was collected, and the relative transcript level of the target mRNA in each sample was calculated by normalization of Ct values to the reference mRNA (GAPDH). Primer sequences used for RT-PCR are as shown in Table 1.

## Gα protein pulldown assay

BL21 *E.coli* cells were transformed with constructs expressing GST-tagged GINIP protein or GST-tagged GRK2 N-terminal domain. Cells were incubated in terrific broth (http://cshprotocols.cshlp.org/content/2015/9/pdb.rec085894.full), and protein expression induced by the addition of 0.5 µg/ml IPTG. Cells were collected after 8 h, lysed and GST-proteins were collected on Glutathione-conjugated beads. Beads were washed four times, eluted with 50 mM glutathione, and proteins desalted on a gel filtration column. Aliquots of 20 µg aliquots were stored frozen and thawed shortly before use.

For pulldown assays, HUVECs infected with lentivirus encoding GluGlu-tagged Gi mutant or lentivirus encoding both GluGlu-tagged Gq mutant and Ric8A were lysed in cell lysis buffer composed of 10 mM Tris-HCl pH 7.5, 150 mM NaCl, 1% Triton X-100, 5 mM DTT and additional 2 µg/ml of GST-GINIP or GST-GRK2N, and the lysates were incubated for 10 min at 4 °C with gentle agitation with glutathione beads. Beads were washed three times with cell lysis buffer, isolated proteins were solubilized in SDS sample Laemmle buffer, and were analyzed by western blotting.

## Mass spectrometry screening for Gα-binding receptors

To create a Gα protein variant that interacts more strongly with GPCR, a green fluorescent protein (GFP) tag was inserted for affinity purification, alongside a mutation of four alanine residues into helix α5 (ins4A), as this hinders GTP binding and maintains the association with GPCRs even in the presence of GTP. In addition, for the extraction of GPCRs while preserving their configuration and lipid environment, a non-detergent nanodisc approach utilizing styrene-maleimide anhydride (SMA) copolymers was employed according to protocols described previously.

In the co-immunoprecipitation process, GFP-Trap® nanobody beads were utilized to capture the interactions between GFP-tagged Gα protein variants and activated designer receptors exclusively activated by designer drugs (DREADDs). The resulting complexes were subjected to subsequent proteomic analysis to determine which GPCRs interacted preferentially with the Gi1(Q306K) variant following the induction of fluid shear stress. Identified receptors were further analyzed by western blotting as described in the supplementary material.

## Measurement of Golgi orientation

Following exposure to laminar shear stress, HUVECs were washed with PBS, fixed for 10 min with 3.7% PFA in PBS, permeabilized with 0.5% Triton X-100 in PBS for 10 min, blocked with 3% BSA in PBS for 30 min and then incubated with GM130 antibody (1:500, BD bioscience) at 4 °C for overnight. After washing, cells were incubated with Hoechst and Alexa647-conjugated secondary antibodies at RT for 1 h to visualize the nuclei and Golgi apparatus, respectively. Images were captured with ×20 objective on a Leica Sp8 confocal microscope with the Leica Application Suite (LAS) software.

Golgi polarization was calculated as a vector that connects the center of mass of the nucleus to the center of mass of the Golgi.

**Table 1.  Primer sequences used for quantitative PCR in this study.**

| Gene name | Forward (5'-3') | Reverse (5'-3') |
| --- | --- | --- |
| Human LPHN1 | CAGTACGACTGTGTCCCCTACA | GCACCATGCGCCAGACTG |
| Human LPHN2 | GTCCAATATGAATGTGTCCCTTACA | GCACCAAGCACCCGCCTT |
| Human LPHN3 | GTGCAGTATGAATGTGTCCCTTACA | GCACCACGCCCCAGATTG |
| Human KLF2 | CGGCAAGACCTACACCAAGA | TGGTAGGGCTTCTCACCTGT |
| Human GAPDH | TGCACCACCAACTGCTTAGC | GGCATGGACTGTGGTCATGAG |
| zebrafish LPHN1a | CAGTATGAATGTGTGCCATACA | ACACCATGCCCCCGACTG |
| zebrafish LPHN2a | GTCCAGTATGAATGTGTTCCCTACA | GCACCACGACCCCGCTTG |
| zebrafish LPHN2b.1 | GTCCAGTATGAGTGTGTGCCATACA | TTGCACCAAGCTCCACTC |
| zebrafish LPHN3.1 | GTTCAGTATGAGTGTGTGCCATACA | ACACCATGATCCAGCCTG |
| zebrafish beta-actin | GATCTGGCATCACACCTTCTAC | TCTTCTCTCTGTTGGCTTTGG |

The angle between this vector and the direction of flow was then determined, with 0° defined as against the flow. Oriana V4 software was used to calculate the angle mean and circular SD and to generate a rose plot for data visualization.

## In vitro permeability assay using the FITC–streptavidin/biotinylated-fibronectin

Slides/dishes were coated with biotinylated-fibronectin at 0.25 mg/mL overnight at 4 °C. HUVECs were seeded of $2 \times 10^6$ cells per well on glass-bottom 6-well plates. Cells were subjected to laminar shear stress by orbital shaking at 150 rpm for 24 h. Immediately after the cessation of flow, FITC–Streptavidin (1:1000, Invitrogen) was added to the cells for 1 min before the fixation by the addition of 3.7% (v/v) paraformaldehyde. Following subsequent washing and blocking, Alexa Fluor 488 Phalloidin (1:200; Invitrogen) was added to stain F-actin filaments. Images were captured from five random fields, and FITC–streptavidin staining was quantified using ImageJ.

## In vitro endothelial 3D invasion assay under shear stress

In order to equalize the VEGF concentration in the collagen gel and endothelial culture medium, VEGF-189 (R&D systems, 8147-VE), a matrix-binding isoform of VEGF, was applied to the bottom of each well at a concentration of 50 ng/ml prior to adding collagen matrices. Rat tail collagen type I (Thermofisher Scientific, A1048301) was employed to prepare collagen matrices at the concentration of 1.5 mg/ml with additional NaOH to facilitate the collagen matrix assembly by neutral pH. After thorough mixing, 1 ml of collagen I solution was placed in each well of a 12-well plate (Corning #3513) and polymerized at 37 °C and 5% $CO_2$ for 2 h to overnight.

To prepare endothelial cells, confluent HUVEC culture plates were washed with PBS, trypsinized, and counted. After pelleting, cell pellets were resuspended in EGM-2 medium (Lonza, #CC-3162) free of VEGF and heparin to establish a positive gradient of VEGF from cells to the bottom of the collagen gel, and 300,000 cells per well were placed on the collagen gel and incubated for 24 h. Following incubation, plates were placed on an orbital shaker and ECs were exposed to orbital shaking at 150 rpm for 48 h as mentioned above in shear stress section. Cells were then fixed in 4% PFA and permeabilized in PBS containing 0.5% Triton X-100. Nonspecific binding was blocked by incubation with PBS containing 3% BSA. Thereafter, cells were subjected to Alexa488-conjugated phalloidin (1:400 dilution; Thermofisher scientific: A12379) and Hoechst (1:1000 dilution; Invitrogen: H1399) for

2 h to stain actin cytoskeleton and cell nuclei, respectively. All images were captured with the Nikon Sp8 confocal microscope and segregated into peripheral region (with uniaxial pulsatile shear) vs. static controls.

## Zebrafish husbandry and handling

Zebrafish were housed and maintained at 28.5 °C in accordance with standard methods approved by the Yale University Institutional Animal Care and Use Committee (#2017-11473) (Lawrence, 2016).

## Generation of LPHN/adgrl knockdown zebrafish using CRISPR/Cas9 ribonucleoproteins

Four zebrafish latrophilin paralogs were identified—*adgrl1a, adgrl2a, adgrl2b.1*, and *adgrl3.1*. Fused sgRNAs were generated targeting each paralog (see Table 2) by annealing locus-specific oligonucleotides to a common 5' universal oligonucleotide and performing in vitro RNA transcription (AmpliScribe T7-Flash Kit, Lucigen, ASF3507) (Gagnon et al, 2014). An injection mix of 50 ng/μL each sgRNA, an equivalent concentration of Cas9 protein (TrueCut Cas9, Invitrogen, A36497), and either 1 μg/μL *silent heart/tnnt2a* (Sehnert et al, 2002) or standard control morpholino (GeneTools) was prepared. In total, 1 nL of the mixture was injected into *Tg(kdrl:ras-mCherry, fli1a:nls-GFP)* zebrafish at the one-cell stage (Parker Hannifin, Picospritzer III). CRISPR reagent efficacy was confirmed by T7 endonuclease assay (see Appendix Fig. S9). Guides against adgrl2a and 2b.1 were combined to target all LPHN2 isoforms.

## Immunostaining and imaging of ECs in the knockdown zebrafish

CRISPR/Cas9 RNP-injected embryos were either fixed at 48 hpf in 4% paraformaldehyde (Santa Cruz Biotechnology, SC-253236) and immunostained for GFP (1:300 chicken anti-GFP, abcam, ab13970), mCherry (1:300 rabbit anti-mCherry, abcam, ab167453), and ZO-1 (1:200 mouse anti-ZO-1, Invitrogen, 61-7300) with species-appropriate secondary antibodies (1:400 Invitrogen anti-chicken/-rabbit/-mouse Alexa 488/546/647, A-11039/A- 10040, A-31571) or imaged live. For fixed imaging, the larvae were processed as previously described (Sugden et al, 2017). For live imaging, larvae were anesthetized by immersion in 600 μM tricaine methylsulfonate (Western Chemical, TRS5), and embedded in 1.5% low-melt agarose (Bio-Rad, 1613106). For staining in detail, larvae were fixed overnight in 4% PFA at +4 °C and then

**Table 2. Sequences of sgRNAs and morpholinos used in this study.**

| | Sequence | Target | Description |
|---|---|---|---|
| 5' | GCTGTACGAGTATGCTTCATGGG | *adgrl1a* | sgRNA targeting adgrl1a exon 4 |
| 5' | CGACGTATAAACTACCCCATCGG | *adgrl2a* | sgRNA targeting adgrl2a exon 4 |
| 5' | CCGTCTATGATAAACGCTCCGCC | *adgrl2b.1* | sgRNA targeting adgrl2b.1 exon 4 |
| 5' | AGTCCACGGCTGCGATGTATTGG | *adgrl3.1* | sgRNA targeting adgrl3.1 exon 4 |
| 5' | CATGTTTGCTCTGATCTGACACGCA | *tnnt2a* | morpholino targeting tnnt2a translation |
| 5' | CCTCTTACCTCAGTTACAATTTATA | none | control morpholino |

transitioned into 100% methanol via a series of dilutions. Once in methanol, the larvae were stored at −20 °C overnight, then rehydrated the next day via the same dilution series. Larvae were permeabilized using incubation in acetone for 15 min at −20 °C and then by incubation in 20 μg/mL proteinase K for 20 min at 37 °C. Larvae were refined in 4% PFA for 15 min at room temperature and then incubated in blocking buffer (0.5% Triton X-100, 10% goat serum, 1% BSA, 0.01% sodium azide, and 1% DMSO in PBS) for 2 h at room temperature. Larvae were incubated in primary antibody (1:400 chicken anti-GFP, abcam ab13970) overnight in blocking buffer at +4 °C. Larvae were thoroughly washed in PBS-Tx and incubated overnight at +4 °C in a secondary antibody (1:400 goat anti-chicken Alexa488 A-11039, Thermo-Fisher) in blocking buffer. Following final washes in PBS-Tx, larvae were mounted and imaged. Intersegmental vessel and dorsal aorta diameter were then calculated using ImageJ. All imaging was performed using a Leica SP8 confocal microscope.

## Immunostaining and imaging of mouse aorta

Aortae were perfusion-fixed in situ with 4% paraformaldehyde (overnight) prior to staining with primary beta-catenin antibody (Cell signaling: 9562) and fluorophore-conjugated secondary antibody (Invitrogen, donkey anti-Rabbit IgG (H + L) Highly Cross-Adsorbed Secondary Antibody, Alexa Fluor 647, A31573) in in Claudio buffer (1% FBS, 3% BSA, 0.5% Triton X-100, 0.01% Sodium deoxycholate, 0.02% sodium azide in PBS pH 7.4) (Franco et al, 2013). Cells were stained with Hoechst (Invitrogen: H1399) to label nuclei. Images of stained vessels were acquired using a Leica SP8 confocal microscope with the Leica Application Suite (LAS) software. Endothelial cell alignment in aorta was quantified by measuring the eccentricity of the immunostaining of beta-catenin of the endothelial cells (>150 cells per field of view).

## Extravasation of Evans blue dye for aortic vascular permeability

For aortic vascular permeability measurements, Evans blue dye (20 mg/ml in saline, 2.5 μl/g mouse body weight) (Sigma E2129) was injected retro-orbitally to LPHN2 endothelial cell-specific knockout and their control littermates at 8 weeks old. Thirty minutes post injection, animals were euthanized by an overdose of isoflurane and tissue samples including aorta were harvested and washed with PBS to remove extra fatty tissues. The dissected aorta was prepared en face and imaged with an Olympus stereo zoom microscope. To quantify the blue color associated with extravasation of Evans blue dye, the original RGB images were converted into HSV color model, and the proportion of the blue regions was calculated as the ratio of pixels with hue angles of 120–240° to the entire aorta.

## Immunostaining and imaging of mouse muscle capillaries

Rectus femoris muscle from murine thigh was perfusion-fixed with 4% paraformaldehyde (overnight), and the middle part of the muscle were cryo-sectioned following embedding into OCT compound and stained with primary antibodies for SMA (fluorophore-conjugated) and for CD31 followed by fluorophore-conjugated secondary antibody (Invitrogen, donkey anti-Rabbit

IgG (H + L) Highly Cross-Adsorbed Secondary Antibody, Alexa Fluor 647, A31573) in PBS containing 5% BSA. Hoechst (Invitrogen: H1399) was used to nuclear identification. Images of stained vessels were acquired using a Leica SP8 confocal microscope with the Leica Application Suite (LAS) software.

## Hindlimb ischemia model

Mice were anesthetized and operated with a mixture of 1.5–2% isoflurane. The front hair of the right thigh was removed with Nair lotion and the local surgical area was disinfected. A 5-mm longitudinal skin incision was made in the right thigh. The distal femoral artery, proximal to the popliteal artery and saphenous artery, was explored and dissected. In all, 6-0 silk sutures (Syneture) were double used for the ligation before excision of the arterial vessel bed between the distal end of the superficial epigastric artery and the trifurcation of the femoral artery into the descending genicular, popliteal, and saphenous branches. The ligated artery was cut between two sutures. The venous structures and accompanying peripheral nerves were kept intact. The overlying skin was closed with a 6-0 prolene (Ethicon) and disinfected. At the end of the procedure, Buprenorphine was administered subcutaneously at 0.05 mg/kg, once pre-emptively, then every 12 h for 72 h. The survival rate for this surgery was 100%.

## Laser Doppler perfusion imaging

Hindlimb perfusion was assessed noninvasively in the plantar foot before, 1, 3, 7, 14, and 21 days after hindlimb ischemia by scanning laser-Doppler [model LDI2-IR modified for high resolution; Moor Instruments, Wilmington, DE]. The hindquarters were placed on the top of a heater pad during scanning to minimize variation in body temperature. Doppler perfusion of the plantar foot was assessed within anatomically defined regions of interest (ROIs; Moor Instruments). The ROI for the plantar foot consisted of the hind paw margins. Procedures for minimal preparation, scanning, and ROI selections have been described in detail previously (Zhuang et al, 2011). Low or no perfusion is displayed as dark blue, whereas the highest degree of perfusion is displayed as red. These images were quantitatively converted into histograms that represented the amount of blood flow on the x-axis and the number of pixels on the $y$ axis in the traced area. The average blood flow in each histogram was calculated and the LDI index was determined as the ratio of ischemic to non-ischemic hindlimb blood perfusion.

## Volumetric micro-CT angiogram for quantitative arteriogenesis

After the mice were euthanized, the vasculature was flushed with 0.9% normal saline containing heparin (1000 IU/L) for 3 min, followed by 4% paraformaldehyde for 5 min at 100 mm Hg pressure. Fresh homemade 20% bismuth nanoparticles mixed in 5% gelatin were used as a micro-CT contrast agent and injected over 2 min with a syringe pump, as described before (Tirziu et al, 2005). The mice were then immediately chilled on ice for more than 30 min and immersion-fixed in 2% paraformaldehyde overnight. The peripheral vasculature in the ischemic hindlimb and the contralateral normal hindlimb was imaged with a high-resolution micro-CT imaging system (GE eXplore Locus SP),

set to a 0.007-mm effective detector pixel size. Microview software (GE Healthcare) was used for initial reconstruction. Advanced workstation with various software was used for 3D reconstruction and segmentation of the whole hindlimb at the 21 μm spatial resolution. Using the knee as the breakpoint, we re-sliced the 3D vasculature into 200 slices both in the thigh and in the calf regions. Quantification was performed by use of a NIH Image. The data were expressed as vascular segment numbers, representing the total number of vessels, of specified diameter, counted in 200 *z*-sections for the thigh region and another 200 *z*-sections for the calf region.

### Mouse treadmill fatigue test

Mice were first acclimatized by staying on the motionless treadmill machine for 30 min on day 1 and then trained on the treadmill moving at 12 m/min for 30 min on day 2. Fatigue tests were then conducted on days 3 and 4. Mice were placed on the treadmill and the shock rod was activated. Treadmill speeds were increased as follows: 0 m/min, for 12 min to measure basal metabolism, 12 m/min, for 1 min, then increasing at a rate of +1 m/min each 1 min until exhaustion. Exhaustion (endpoint of treadmill cessation) was defined as the point at which mice continuously contacted with the shock rod for 5 continuous seconds or when mice stopped running three times within 5 s. Before each testing session, Oxymax software (Columbus Instruments, Columbus, OH, USA) and open circuit indirect calorimetry treadmills (Metabolic Modular Treadmill, Columbus Instruments, Columbus, OH) were calibrated and checked for hardware malfunctions according to the manufacturer's instructions.

## Data availability

The datasets produced in this study are available in the following databases: All data: BioStudies S-BIAD928.

The source data of this paper are collected in the following database record: biostudies:S-SCDT-10_1038-S44318-024-00142-0.

## Peer review information

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

## Acknowledgements

This work was supported by NIH grants RO1 HL75092 and R01 HL151469 to MAS, NHLBI 5R01HL130246-05 to SN, and the Uehara Memorial Foundation postdoctoral fellowship and MEXT | Japan Society for the Promotion of Science (JSPS) Overseas Research Fellowships (2014, receipt number: 805) to KT. We thank Hamit Harun Dag (Istanbul University) for help of synthesizing SMA polymer; Shahid Mansuri (Yale University), Jeremy Herrera (University of Manchester) and David Knight (University of Manchester) for expert advice for mass spectrometry experiments; Didier Trono (École polytechnique fédérale de Lausanne) for lentiviral packaging plasmids pMD2.G (Addgene plasmid # 12259; http://n2t.net/addgene:12259; RRID:Addgene_12259) and psPAX2 (Addgene plasmid # 12260; http://n2t.net/addgene:12260; RRID:Addgene_12260); Bryan Roth (University of North Carolina) for pcDNA5/FRT-HA-hM3D(Gq) (Addgene plasmid # 45547; http://n2t.net/addgene:45547; RRID:Addgene_45547) and pcDNA5/FRT-HA-hM4D(Gi)(Addgene plasmid # 45548; http://n2t.net/addgene:45548; RRID:Addgene_45548); Aziz Moqrich (Institut de Biologie du Dévelopment de Marseille) for GST-GINIP plasmid. We are grateful to Peter Newman (Medical College of Wisconsin) for the kind gift of PECAM-1 antibody and helpful discussions. Schematic models are inspired by BioRender templates (2022).

## Author contributions

**Keiichiro Tanaka**: Conceptualization; Data curation; Software; Formal analysis; Supervision; Validation; Investigation; Visualization; Methodology; Writing—original draft; Project administration; Writing—review and editing. **Minghao Chen**: Formal analysis; Investigation; Methodology. **Andrew Prendergast**: Investigation; Methodology. **Zhenwu Zhuang**: Investigation; Methodology. **Ali Nasiri**: Investigation; Methodology. **Divyesh Joshi**: Methodology. **Jared Hintzen**: Methodology. **Minhwan Chung**: Methodology. **Abhishek Kumar**: Methodology. **Arya Mani**: Supervision. **Anthony Koleske**: Supervision. **Jason Crawford**: Supervision. **Stefania Nicoli**: Supervision. **Martin A Schwartz**: Conceptualization; Supervision; Funding acquisition; Writing—original draft; Writing—review and editing.

Source data underlying figure panels in this paper may have individual authorship assigned. Where available, figure panel/source data authorship is listed in the following database record: biostudies:S-SCDT-10_1038-S44318-024-00142-0.

## Disclosure and competing interests statement

The authors declare no competing interests.

