## [Peer Review File · The EMBO Journal]

Latrophilin-2 mediates fluid shear stress mechanotransduction at endothelial cell-cell junctions

Keiichiro Tanaka, Minghao Chen, Andrew Prendergast, Zhen Zhuang, Ali Nasiri, Divyesh Joshi, Jared Hintzen, Minhwan Chung, Abhishek Kumar, Arya Mani, Anthony J. Koleske, Jason Crawford, Stefania Nicoli, and Martin Schwartz

Corresponding authors: Martin Schwartz (martin.schwartz@yale.edu) , Keiichiro Tanaka (keiichiro.tanaka@yale.edu)

Review Timeline:

Submission Date:	5th Apr 23
Editorial Decision:	16th May 23
Revision Received:	6th Nov 23
Editorial Decision:	10th Jan 24
Revision Received:	20th Feb 24
Accepted:	13th May 24

Editor: Ieva Gailite

Transaction Report:

Dear Dr. Schwartz,

Thank you for submitting your manuscript for consideration by the EMBO Journal. I apologise for the protracted assessment process due to delays in referee report submission. We have now received comments from three reviewers, which are included below for your information.

As you will see from the reports, all reviewers find the proposed role of latrophilins in shear stress-dependent regulation of vascular development of interest. However, they also raise several points that would need to be addressed, including the role of LPHN2 in arterial vs venous vessels and the timescale of LPHN2 interaction with Ga proteins upon fluid flow response. According to the EMBO Press guidelines, and as also raised by reviewers #2 and #3, the proteomics dataset should be uploaded in a publicly accessible database and referee access provided upon revision. From the editorial side, further investigation of the role of S1PR family members (referee #2, point 5) will not be needed.

Based on the interest expressed in the reports, I invite you to address these issues in a revised version of the manuscript. I think it would be helpful to discuss the revision in more detail via email or phone/videoconferencing - please let me know which option you prefer. I should also add that it is The EMBO Journal policy to allow only a single major round of revision and that it is therefore important to resolve the main concerns at this stage.

We generally allow three months as standard revision time, which can be extended to six months in the case of major revisions. As a matter of policy, competing manuscripts published during this period will not negatively impact on our assessment of the conceptual advance presented by your study. However, please contact me as soon as possible upon publication of any related work to discuss the appropriate course of action. Should you foresee a problem in meeting this deadline, please let us know in advance to discuss an extension.

When preparing your letter of response to the referees' comments, please bear in mind that this will form part of the Review Process File and will therefore be available online to the community. For more details on our Transparent Editorial Process, please visit our website: <https://www.embopress.org/page/journal/14602075/authorguide#transparentprocess>. Please also see the attached instructions for further guidelines on preparation of the revised manuscript.

Please feel free to contact me if you have any further questions regarding the revision. Thank you for the opportunity to consider your work for publication. I look forward to discussing your revision.

Yours sincerely,

Ieva Gailite

We realize that it is difficult to revise to a specific deadline. In the interest of protecting the conceptual advance provided by the work, we recommend a revision within 3 months (14th Aug 2023). Please discuss the revision progress ahead of this time with the editor if you require more time to complete the revisions.

Referee #1:

The manuscript by Tanaka et al describes the role of an adhesion-based GPCR (latrophilin2, LPHN2) in endothelial cell mechanotransduction and blood vessel function. Starting from the premise that an unidentified sensor is upstream of PECAM1/VEcadherin/VEGFR/PlexinD1 complex, the authors systematically examine G protein requirements for early signaling and alignment to laminar flow and develop an activation assay to define G protein requirements. They then biochemically link to LPHN2 and show that it is required for flow signaling and alignment upstream of PECAM. They go on to examine mutants in fish and zebrafish to show flow-mediated effects on alignment and permeability in vivo with LPHN2 loss, and challenge mice with ischemia and treadmill to show underlying physiological and pathological effects of LPHN2 loss. They also provide evidence for flow-mediated sprouting defects using microfluidic flow devices, and describe human genetic data linking Adgrl2 to cardiovascular disease. Taken together this study provides ample rigorous evidence for an endothelial mechanosensor upstream of the PECAM complex. The data is well-controlled and complementary and strongly supports the conclusions. These findings have potential to strongly impact the field, as endothelial flow-mediated responses are crucial for proper vascular development and remodeling, and knowing the players and how they intersect with the other components as well as its position at the top of the hierarchy, is an important advance in the field.

There are some points for clarification:

1. Although the mutations (Fig 3I) that might have given some insight to how LPHN2 is flow activated were not informative, would the authors be able to speculate a bit in the discussion as to how LPHN2 might be activated by flow and how that information might be transduced to a signal that intersects with PECAM1?
2. It's interesting that LPHN2 may confer flow sensitivity on pathways that also transduce non-flow signals. From that perspective could the authors consider whether LPHN2 signaling might be upstream of other mechanosensory complexes in endothelial cells as a discussion point?

Minor points:

1. Fig 1 panel K-L - the graph normalizes each control to 1 which masks the high baseline activity of Gi2EE lanes - suggest normalize all lanes to first control.
2. Fig 2 panel E - blot quality for pulldown not high quality - at least lanes 1 and 4 do not have consistent signal horizontally. Please consider another example.
3. Fig 4 I-J - Scoring permeability defects in vivo is challenging - the signal is subtle in the mutant aorta. Is there a better example and/or another vascular bed that might reveal a permeability defect more robustly?

Referee #2:

Summary

Tanaka et al. identify in this manuscript latrophilin 1-3 (LPHN/ADGRL1-3) as flow-dependent mechanosensitive G-protein-coupled receptors (GPCRs) that activate G α proteins downstream of shear stress in endothelial cells. The authors used sophisticated biochemical approaches to identify relevant G α proteins and upstream receptors mediating flow-dependent elongation of endothelial cells and activation of Src signalling. They further demonstrate, using zebrafish and mouse models, that loss of LPHN2 in vivo decreased the elongation of endothelial cells, impaired endothelial cell junction maturation and decreased angiogenic capacity. Functionally, LPHN2 endothelial-specific KO decreased recovery in the hindlimb ischemia model and increased permeability in the mouse aorta.

In general, this manuscript provides a novel perspective on how shear stress modulates endothelial cell signalling. The major novelty resides in the identification of LPHN2 as a key flow-dependent mechanotransducer. The role of LPHN2 as an important regulator of angiogenesis (10.1083/jcb.202006033), the link between LPHN and G α proteins (10.1515/bmc-2014-0032), or the link between G α proteins and flow sensing (10.1113/jphysiol.2009.172643, 10.1073/pnas.060722410) have all been previously shown in previous publications. One additional relevant information is placing GPCR-signaling upstream of the mechanosensitive PECAM1-VEGFR2-VE-cadherin complex, as well as the detailed analysis of the vascular phenotypes of LPHN2 endothelial-specific KO. However, there are several aspects that would benefit from additional control experiments and additional information.

Major comments

1 - Incorrect/Poor reporting of materials and methods

The authors should specify which G α genes were targeted in this study. As literature is sometimes confusing and there are multiple members for each of the Gi and Gq/11 family of proteins, clarification of this aspect would be very informative. For instance, when authors discuss the knockdown of Gq/11 are they referring to which specific gene? GNAQ, GNA11, GNA14, or GNA15? Or are authors targeting all of them at once? Similarly, the authors have hidden the siRNAs used in this study by writing in the methods "G α protein siRNAs were as described previously⁴⁸⁻⁵⁰." All reagents should be properly reported here with their corresponding catalogue numbers. Intriguingly, the reference for the antibody used to probe for Gq/11 is incorrect, as no catalogue #26060 exists in NewEast biosciences site. Alongside, no description of the cDNA used to perform rescue experiments for Gi or Gq proteins is provided.

Another example is the mysterious "Claudio buffer" for immunostaining aortas. What is Claudio's buffer composition, is there any reference for its origin?

Shear stress conditions should also be better reported in material and methods.

Those are a few examples of inadequate reporting of methods and materials. The authors should carefully revise this section and report the details in full.

2- Mechanosensing differences between arterial and venous cells.

The majority of the experiments performed by the authors were done using HUVECs, which are of vein origin. Yet, most of the phenotypes reported in vivo are seen in arteries. Can the authors provide further evidence in other vascular beds? Given the well-reported endothelial cell identity differences between arterial and venous cells, the authors should report if veins and arteries differ in the expression levels of the key molecules involved in this study. Moreover, are defects of endothelial cell morphology and permeability in LPHN2 KD/KO animals restricted to the aorta? Finally, the authors could also perform a few confirmatory experiments in vitro using an arterial cell line, such as HAECs, or similar.

3- Confirmation that Gi2 K307 is the key functional amino acid.

The authors state that the (Gi2) K307 residue "distinguishes the G α subunits participate in flow signaling", including in Gq/11. The reviewer appreciates the elegant rescue experiments; however, the authors do not provide any evidence that Gq/11 has the same pattern of residues. The authors should show the sequence similarity in the Gq protein. In addition, the authors should demonstrate that expression of mutated Gi2 K307Q fails to rescue Gi/Gq/11 double knockdowns. Similarly, it would be interesting to analyse if Gi2 K307Q co-immunoprecipitates with LPHN2 or not.

4- Timescale of signalling and biological events.

The authors reported that shear stress activates LPHN2 and G α proteins within 30sec-2mins. However, flow-dependent elongation of endothelial cells takes hours. Is LPHN2-G α activation sustained over time, or is it a transitory state, or shows oscillatory behaviour? Moreover, the authors reported that LPHN2 directly/indirectly binds to PECAM1. Does LPHN2 and/or Gi or Gq/11 also pulldowns VEGFR2 or VE-cadherin, or is this restricted to PECAM1? Finally, it has been widely reported that shear stress not only promotes endothelial cell elongation but also polarization against the flow direction. Does LPHN2-G α signalling affect endothelial cell polarity?

5- Proteomics would benefit from better reporting and justification.

Given that the authors did a very large effort using mass-spec to identify proteins that were pulldown with G α proteins, it is recommended to provide the entire list of proteins identified and to deposit the mass-spec data in a suitable repository. Moreover, the scientific justification to investigate the role of LPHN2 in this context was not clear. Both S1PR1 and ADGRL3 (LPHN3) were the prime targets identified in the study, yet the authors justified focusing on LPHNs based on negative data on S1PR1. Yet, afterwards, the authors focused on a different family member (LPHN2) instead of the initial target (LPHN3), as

LPHN3 did not show a phenotype when knocked-down. Thus, the same principle of looking for other members of the LPHN family could be applied to S1PR1. To further clarify this aspect, the authors should investigate the effect of knocking down S1PR family members.

6- Genetic association of LPHN2 and human disease is weak.

The genetic link between LPHN2 and human disease is interesting but very speculative at this stage. As the authors do not provide any further information on how those SNPs may affect LPHN2 expression or function, nor the human pathologies (increased atherosclerosis or hypertension) are directly linked to any phenotype described in the manuscript. Thus, the association between the role of LPHN2 and human cardiovascular pathology is weak. The reviewer recommends that the authors either provide further information or that claims should be substantially tone-down in the manuscript.

Minor comments

1- What is the role of LPHN2 EC-KO in retinal development?

2- The numbering of appendixes is not linear, for instance, Appendix fig. S2D, S2E come very late in the text after Appendixes 3 and 4 have been discussed. Same thing for Appendix fig. S7 that is cited ahead of Appendix fig. S5 or 6. This complicates the interpretation of figures when reading the manuscript.

3- The study would benefit from an extended introduction to the mechanisms of activation of G α proteins and their family members.

4- There are several grammatical mistakes throughout the manuscript that should be corrected.

5- Line 90 - Gi1(N167D) instead of Gi1 (N166D).

6- In some figures are missing the staining legends and scale bar.

7- The reviewer recommends putting all antibodies (primary and secondary) in a table as a list with the respective information, making it easier to identify them.

Referee #3:

In this manuscript by Tanaka et al, the authors demonstrate that latrophilins mediate shear stress sensing of endothelial cells. They show that laminar flow activates G α /11 and Gi2, which is needed for the endothelial cells to align in the direction of flow. They show that LPHN2 can bind to activated Gi1 in response to flow, an interaction that is dependent on the K306 residue. By using LPHN2 knockout models the authors further demonstrate that LPHN2 is needed for vascularization in various tissues. The study strengths are that experiments are well controlled and span experiments that address biochemical interactions and activations, cellular remodeling and the functional importance of LPHN2 for blood vessels in vivo. The concept that LPHN2 contributes to flow sensing is novel and an important new insights for the vascular biology field.

There are several comments that the authors may address:

Major:

- It is difficult to grasp the timescale of events. Endothelial cell alignment to flow is a longterm process compare to the rapid activation of GPCRs. It is therefore important to establish how long the interaction of G α proteins with LPHN2 at the junctional membrane is active. Does the complex needs constitutive activation? Or is it a temporary response?
- The authors study the onset of laminar flow on endothelial cells. In their in vivo models examples are shown of the dorsal aorta of zebrafish, and in the mice of the adult aorta. both models can be employed to study the endothelium during vascular remodeling at the onset, and following established levels, of flow (for instance in the ISVs), and to investigate the retinas of the mice to be able to investigate the functional importance of LPHN2 on different vascular beds (and flow areas) of the microcirculation. Also the disturbed and laminar flow areas in the aortic arch are excellent tissues to study whether LPHN2 specifically affects the endothelium at distinct flow areas.
- Does LPHN2 bind to VE-cadherin in response to flow?
- Given the prominent effects of Gi/LPHN2 on endothelial junctions, and less blue staining in LPHN2 aortas, what is the effect of LPHN2 depletion on endothelial barrier function under flow?
- What is the integrity of the arteries and veins in the knockout zebrafish? Are they normally perfused and what is their permeability?
- Raw data from proteomics, where the interaction of LPHN2 with G α proteins was discovered, do not seem to be in the paper.

Minor:

- How does this complex relate to flow induced Piezo activation? Are these mutually dependent first flow responding pathways? More elaborate discussion of the integration of this study and the earlier results from Julián Albarrán-Juárez et al (JEM 2018) regarding disturbed flow in the endothelial-specific G α /11 mice might help readers.
- The flow-induced orientation graphs miss an indicator explaining what the individual bars represent.
- Do the authors think that the SNPs might modulate the interaction of LPHN2 with G α proteins?

Referee #1:

The manuscript by Tanaka et al describes the role of an adhesion-based GPCR (latrophilin2, LPHN2) in endothelial cell mechanotransduction and blood vessel function. Starting from the premise that an unidentified sensor is upstream of PECAM1/VEcadherin/VEGFR/PlexinD1 complex, the authors systematically examine G protein requirements for early signaling and alignment to laminar flow and develop an activation assay to define G protein requirements. They then biochemically link to LPHN2 and show that it is required for flow signaling and alignment upstream of PECAM. They go on to examine mutants in fish and zebrafish to show flow-mediated effects on alignment and permeability in vivo with LPHN2 loss, and challenge mice with ischemia and treadmill to show underlying physiological and pathological effects of LPHN2 loss. They also provide evidence for flow-mediated sprouting defects using microfluidic flow devices, and describe human genetic data linking Adgrl2 to cardiovascular disease. Taken together this study provides ample rigorous evidence for an endothelial mechanosensor upstream of the PECAM complex. The data is well-controlled and complementary and strongly supports the conclusions. These findings have potential to strongly impact the field, as endothelial flow-mediated responses are crucial for proper vascular development and remodeling, and knowing the players and how they intersect with the other components as well as its position at the top of the hierarchy, is an important advance in the field.

We sincerely thank the reviewer for this positive evaluation.

There are some points for clarification:

1. Although the mutations (Fig 3I) that might have given some insight to how LPHN2 is flow activated were not informative, would the authors be able to speculate a bit in the discussion as to how LPHN2 might be activated by flow and how that information might be transduced to a signal that intersects with PECAM1?

We have updated the Discussion accordingly. However, we also have some new results that we present in figure 5 suggesting activation of latrophilins through membrane fluidization. If the reviewers agree, we can show these results in the manuscript.

2. It's interesting that LPHN2 may confer flow sensitivity on pathways that also transduce non-flow signals. From that perspective could the authors consider whether LPHN2 signaling might be upstream of other mechanosensory complexes in endothelial cells as a discussion point?

We have evidence about LPHN2 and other flow pathways but the work is incomplete. We prefer not to speculate until the data are ready to present.

Minor points:

1. Fig 1 panel K-L - the graph normalizes each control to 1 which masks the high baseline activity of Gi2EE lanes - suggest normalize all lanes to first control.

We have updated Figure 1L and normalized all lanes to the first control as requested.

2. Fig 2 panel E - blot quality for pulldown not high quality - at least lanes 1 and 4 do not have consistent signal horizontally. Please consider another example.

Thank you for the suggestions. We include a different example of the pulldown assays in Figure 2E.

3. Fig 4 I-J - Scoring permeability defects in vivo is challenging - the signal is subtle in the mutant aorta. Is there a better example and/or another vascular bed that might reveal a permeability defect more robustly?

Given the large amount of work required to discover LPHN2 as a mediator of flow signaling and further mechanistic characterization, a detailed analysis of multiple vascular beds is beyond the scope of the current manuscript. However, to support the notion that latrophilins regulate endothelial permeability, we have added in vitro endothelial permeability assays, which show that latrophilins are required for flow-induced barrier stabilization (Figure 4I), consistent with the requirement for LPHN2 in flow-induced linearization of junctions (Fig 4A,B).

Referee #2:

Summary

Tanaka et al. identify in this manuscript latrophilin 1-3 (LPHN/ADGRL1-3) as flow-dependent mechanosensitive G-protein-coupled receptors (GPCRs) that activate G α proteins downstream of shear stress in endothelial cells.

The authors used sophisticated biochemical approaches to identify relevant G α proteins and upstream receptors mediating flow-dependent elongation of endothelial cells and activation of Src signalling. They further demonstrate, using zebrafish and mouse models, that loss of LPHN2 in vivo decreased the elongation of endothelial cells, impaired endothelial cell junction maturation and decreased angiogenic capacity. Functionally, LPHN2 endothelial-specific KO decreased recovery in the hindlimb ischemia model and increased permeability in the mouse aorta.

In general, this manuscript provides a novel perspective on how shear stress modulates endothelial cell signalling. The major novelty resides in the identification of LPHN2 as a key flow-dependent mechanotransducer. The role of LPHN2 as an important regulator of angiogenesis (10.1083/jcb.202006033), the link between LPHN and G α proteins (10.1515/bmc-2014-0032), or the link between G α proteins and flow sensing (10.1113/jphysiol.2009.172643, 10.1073/pnas.060722410) have all been previously shown in previous publications. One additional relevant information is placing GPCR-signaling upstream of the mechanosensitive PECAM1-VEGFR2-VE-cadherin complex, as well as the detailed analysis of the vascular phenotypes of LPHN2 endothelial-specific KO. However, there are several aspects that would benefit from additional control experiments and additional information.

Major comments

1 – Incorrect/Poor reporting of materials and methods

The authors should specify which G α genes were targeted in this study. As literature is sometimes confusing and there are multiple members for each of the Gi and Gq/11 family of proteins, clarification of this aspect would be very informative. For instance, when authors discuss the knockdown of Gq/11 are they referring to which specific gene? GNAQ, GNA11, GNA14, or GNA15? Or are authors targeting all of them at once? Similarly, the authors have hidden the siRNAs used in this study by writing in the methods "G α protein siRNAs were as described previously"⁴-50." All reagents should be properly reported here with their corresponding catalogue numbers. Intriguingly, the reference for the antibody used to probe for Gq/11 is incorrect, as no catalogue #26060 exists in NewEast biosciences site. Alongside, no description of the cDNA used to perform rescue experiments for Gi or Gq protein is provided.

Another example is the mysterious "Claudio b'ffer" for immunostaining aortas. What is

Claudio's buffer composition, is there any reference for its origin?
Shear stress conditions should also be better reported in material and methods.
Those are a few examples of inadequate reporting of methods and materials. The authors should carefully revise this section and report the details in full.

Thank you for the suggestions.

Knockdown of Gq/11 refers to the simultaneous knockdown of the GNAQ gene and the GNA11 gene using an siRNA that targets an identical sequence in GNAQ and GNA11, as reported previously. GNA14/15 expression in HUVECs is almost undetectable, therefore we did not test it in this study. All siRNAs used to target G proteins are custom sequences based on previous publications as referred to in the Method section, thus, do not have catalogue numbers.

The NewEast Gq/11 antibody used in this study has been discontinued and was replaced with a new antibody. However, we still have the PDF file of the datasheet #26060 for reference and can provide it if necessary. We provided the complete cDNA sequences of G proteins as supplemental data 1.

We have added the composition of Claudio buffer and a reference to the original paper in the Methods.

We have revised the Methods to provide more details about experimental conditions, including the shear conditions for the new experiments.

2- Mechanosensing differences between arterial and venous cells.

The majority of the experiments performed by the authors were done using HUVECs, which are of vein origin. Yet, most of the phenotypes reported *in vivo* are seen in arteries. Can the authors provide further evidence in other vascular beds? Given the well-reported endothelial cell identity differences between arterial and venous cells, the authors should report if veins and arteries differ in the expression levels of the key molecules involved in this study. Moreover, are defects of endothelial cell morphology and permeability in LPHN2 KD/KO animals restricted to the aorta? Finally, the authors could also perform a few confirmatory experiments *in vitro* using an arterial cell line, such as HAECs, or similar.

We have added Figure S7 to confirm a similar requirement for LPHN2 in HAECs. This is unsurprising as HUVECs and HAECs respond similarly to fluid shear stress (PMID 33468662, 14718748). We generally feel that this issue is better addressed *in vivo*. Figure 6C shows that LPHN2 deletion mainly affects arteries in zebrafish. Additionally, to assess expression levels between arterial and venous ECs, we have checked publicly available RNAseq data (GSE131681) and compare latrophilin expression in endothelial subpopulations, shown in supplemental figure S5A. Figures 6F-I demonstrate the impact of LPHN2 in femoral arteries. Together, these data provide extensive evidence that LPHN2 is important in arterial flow sensing.

3- Confirmation that Gi2 K307 is the key functional amino acid.

The authors state that the (Gi2) K307 residue "distinguishes the G α subunits participate in flow signaling", including in Gq/11. The reviewer appreciates the elegant rescue experiments;

however, the authors do not provide any evidence that Gq/11 has the same pattern of residues. The authors should show the sequence similarity in the Gq protein. In addition, the authors should demonstrate that expression of mutated Gi2 K307Q fails to rescue Gi/Gq/11 double knockdowns. Similarly, it would be interesting to analyse if Gi2 K307Q co-immunoprecipitates with LPHN2 or not.

We focused on the amino acids where Gi2 is similarly to Gq and G11 but different from Gi1 and Gi2. There are only 2 such residues, 166 and 307, shown in Figure 1E, which gives the sequence across all the relevant isoforms. We have now added a new experiment where we performed the rescue experiment with Gi2 K307Q and found that this mutant could not rescue the alignment of Gi/Gq/11 double knockdowns (Appendix fig S1G and Appendix fig S2D).

4- Timescale of signalling and biological events.

The authors reported that shear stress activates LPHN2 and G α proteins within 30sec-2mins. However, flow-dependent elongation of endothelial cells takes hours. Is LPHN2-G α activation sustained over time, or is it a transitory state, or shows oscillatory behaviour? Moreover, the authors reported that LPHN2 directly/indirectly binds to PECAM1. Does LPHN2 and/or Gi or Gq/11 also pull-downs VEGFR2 or VE-cadherin, or is this restricted to PECAM1? Finally, it has been widely reported that shear stress not only promotes endothelial cell elongation but also polarization against the flow direction. Does LPHN2-G α signalling affect endothelial cell polarity?

We have found that the binding between Lphn2 and G α remains elevated after 24 hours flow compared to static conditions and have added these data to Figure 3H. In addition to PECAM-1, Lphn2-G α also showed increased association with VE-Cadherin, suggesting that Lphn2 interacts with the known endothelial junctional mechanosensory complex.

Regarding cell polarity, we have now checked the orientation of the Golgi apparatus. We confirmed that Golgi polarized more against the flow direction in control cells, which was blocked by Lphn2 knockdown. These results have been added as Figs S5F-G.

5- Proteomics would benefit from better reporting and justification.

Given that the authors did a very large effort using mass-spec to identify proteins that were pulled down with G α proteins, it is recommended to provide the entire list of proteins identified and to deposit the mass-spec data in a suitable repository.

Moreover, the scientific justification to investigate the role of LPHN2 in this context was not clear. Both S1PR1 and ADGRL3 (LPHN3) were the prime targets identified in the study, yet the authors justified focusing on LPHNs based on negative data on S1PR1. Yet, afterwards, the authors focused on a different family member (LPHN2) instead of the initial target (LPHN3), as LPHN3 did not show a phenotype when knocked-down. Thus, the same principle of looking for other members of the LPHN family could be applied to S1PR1. To further clarify this aspect, the authors should investigate the effect of knocking down S1PR family members.

We uploaded the original mass spec data to the BioStudies data repository (accession number: S-BIAD928). In addition, we knocked down S1PR1 and confirmed that this did not block alignment in the direction as shown in Appendix figure S4F and S4G. Fig 3 shows that knockdown of Lphn1 and 3 did not affect multiple flow responses, as expected from their very low expression in HUVECs. Indeed, LPHN3 is essentially not

expressed. Its presence in the mass spec results is most likely a misidentification. If the reviewer is asking us to knock down other S1PR receptors that were not identified in the proteomic assay, we do not see any rationale for doing so.

6- Genetic association of LPHN2 and human disease is weak.

The genetic link between LPHN2 and human disease is interesting but very speculative at this stage. As the authors do not provide any further information on how those SNPs may affect LPHN2 expression or function, nor the human pathologies (increased atherosclerosis or hypertension) are directly linked to any phenotype described in the manuscript. Thus, the association between the role of LPHN2 and human cardiovascular pathology is weak. The reviewer recommends that the authors either provide further information or that claims should be substantially tone-down in the manuscript.

We have toned down our conclusion here and explained that establishing a functional link will require further analysis of the genetics to determine if the reported SNP is functional or if it is in disequilibrium with another SNP.

Minor comments

1- What is the role of LPHN2 EC-KO in retinal development?

We have some preliminary results regarding retinal development but want to do a detailed developmental analysis. Therefore, this is beyond the scope of the current manuscript.

2- The numbering of appendixes is not linear, for instance, Appendix fig. S2D, S2E come very late in the text after Appendixes 3 and 4 have been discussed. Same thing for Appendix fig. S7 that is cited ahead of Appendix fig. S5 or 6. This complicates the interpretation of figures when reading the manuscript.

We have corrected the order of supplemental figures.

3- The study would benefit from an extended introduction to the mechanisms of activation of Gα proteins and their family members.

Thank you for the suggestion. We have expanded the Introduction concerning activation mechanisms for Gα proteins.

4- There are several grammatical mistakes throughout the manuscript that should be corrected. We have carefully corrected grammatical errors.

5- Line 90 - Gi1(N167D) instead of Gi1 (N166D).

Corrected, thank you.

6- In some figures are missing the staining legends and scale bar.

We have added scale bars and staining information to figure legends.

7- The reviewer recommends putting all antibodies (primary and secondary) in a table as a list with the respective information, making it easier to identify them.

Thank you, we have added a table to the Method section.

Referee #3:

In this manuscript by Tanaka et al, the authors demonstrate that latrophilins mediate shear stress sensing of endothelial cells. They show that laminar flow activates Gαq/11 and Gi2, which is needed for the endothelial cells to align in the direction of flow. They show that LPHN2 can bind to activated Gi1 in response to flow, an interaction that is dependent on the K306 residue. By using LPHN2 knockout models the authors further demonstrate that LPHN2 is needed for vascularization in various tissues. The study strengths are that experiments are well controlled and span experiments that address biochemical interactions and activations, cellular remodeling and the functional importance of LPHN2 for blood vessels in vivo. The concept that LPHN2 contributes to flow sensing is novel and an important new insights for the vascular biology field.

There are several comments that the authors may address:

Major:

- It is difficult to grasp the timescale of events. Endothelial cell alignment to flow is a longterm process compared to the rapid activation of GPCRs. It is therefore important to establish how long the interaction of Gα proteins with LPHN2 at the junctional membrane is active. Does the complex need constitutive activation? Or is it a temporary response?

We have checked the binding between LPHN2-Gα and PECAM1-VE-Cadherin. We confirmed that binding is induced at the onset of LSS and now show that it is maintained for at least 24 hours flow (Figure 3D-3G).

- The authors study the onset of laminar flow on endothelial cells. In their in vivo models examples are shown of the dorsal aorta of zebrafish, and in the mice of the adult aorta. Both models can be employed to study the endothelium during vascular remodeling at the onset, and following established levels, of flow (for instance in the ISVs), and to investigate the retinas of the mice to be able to investigate the functional importance of LPHN2 on different vascular beds (and flow areas) of the microcirculation. Also the disturbed and laminar flow areas in the aortic arch are excellent tissues to study whether LPHN2 specifically affects the endothelium at distinct flow areas.

Given the large amount of work required to discover LPHN2 as a mediator of flow signaling and further mechanistic characterization, a detailed analysis of multiple vascular beds and developmental stages is beyond the scope of the current manuscript. We are quite interested in these questions but feel that they deserve a more thorough analysis than can be done as an addition to this already long manuscript.

- Does LPHN2 bind to VE-cadherin in response to flow?

We have now checked binding between LPHN2-Gα and VE-cadherin. At the onset of FSS, LPHN2-Gα binds to VE-Cadherin (Figure 3I). At later times, the binding was weaker but remained above baseline.

- Given the prominent effects of Gi/LPHN2 on endothelial junctions, and less blue staining in LPHN2 aortas, what is the effect of LPHN2 depletion on endothelial barrier function under flow?

To address this question, we have now measured in vitro permeability with control cells and LPHN2 knockdown cells under static and flow conditions. We find that Lphn2 knockdown has little effect under static conditions but blocks the reduction in permeability by laminar shear stress (Fig 4I,J). These results correspond well with the effect on flow-dependent junction linearization (Fig 4A,B). These results further support

our conclusion that LPHN2 is a key mediator of shear stress responses rather than a general mediator of junctional integrity.

- What is the integrity of the arteries and veins in the knockout zebrafish? Are they normally perfused and what is their permeability?

Measuring permeability in zebrafish is quite difficult and would require development of new methods in our facility. We feel that this work is beyond the scope of the current paper.

- Raw data from proteomics, where the interaction of LPHN2 with Ga proteins was discovered, do not seem to be in the paper.

We have uploaded the original raw data to the mass spec repository.

Minor:

- How does this complex relate to flow induced Piezo activation? Are these mutually dependent first flow responding pathways? More elaborate discussion of the integration of this study and the earlier results from Julián Albarrán-Juárez et al (JEM 2018) regarding disturbed flow in the endothelial-specific Gaq/11 mice might help readers.

We understand that this study opens the door to many new questions but it is impossible for one paper to address all of them. We do not know the relationship between Piezo1 and Lphn2. It seems plausible that they are parallel pathways that operate in different settings. However, the reviewer may be unaware that the study from Albarrán-Juárez has generated controversy, with doubts about key aspects being raised in other labs including ours. But we don't feel it is appropriate to say anything about this until results are conclusive.

- The flow-induced orientation graphs miss an indicator explaining what the individual bars represent.

We include the detailed explanation of flow-induced orientation graphs in the legend of Figure 1A.

- Do the authors think that the SNPs might modulate the interaction of LPHN2 with Ga proteins? This is unlikely, at least not directly. Since the SNPs are intronic, the protein sequence is intact. It is more likely to affect Lphn2 alternative splicing or to be in linkage equilibrium with the functional SNPs that affect the coding sequence. We expand on this point in the Discussion.

Dear Martin,

Thank you for submitting a revised version of your manuscript. I sincerely apologise for the protracted assessment process due to delays in referee comment submission and the post-holiday backlog.

Your study has now been seen by all original referees. Reviewers #1 and #2 find that most of their previous concerns have been addressed and now broadly recommend acceptance of the manuscript, while reviewer #3 indicates remaining concerns, which I would ask you to address with adding the information on mass spectrometry experiments, toning down the conclusions and extending discussion in the final revision.

Please also clarify the issues with numerical duplications in the source data that are highlighted by referee #3. There are also further discrepancies I noted in the source data:

- Fig 1B: p-Src blot does not match the figure.
- I cannot open source data files for Fig 1G-N, 2B, 2D-E, 3H, 3K, 4I, 5C - the downloaded files appear empty.
- Unexplained duplicate values have been found also in source data for figures 3L (please note column G), 5A, 6G (summary tab) and 6OP. I have attached the corresponding files with the detected duplications labelled in red. Please take a look and correct as needed. A brief explanation would be very helpful.

There also remain a few editorial points that need addressing before I can extend acceptance of the manuscript:

1. Please submit up to five keywords.
2. Please check that the funding information is correct and identical both in the manuscript and our online system. The Uehara Memorial Foundation postdoctoral fellowship and JSPS Overseas Research Fellowships (2014, receipt number: 805) are currently missing in our online system.
3. Please add a heading for the Introduction section.
4. Please move "Materials and Methods" section after "Discussion".
5. Please move Table 1 and 2 after the main figure legends.
 1. You can convert up to five Appendix figures into Expanded View figures that are collapsible/expandable online. EV Figures should be cited as 'Figure EV1, Figure EV2' etc. in the text and their respective legends should be included in the main text after the legends of regular figures. Further information on the format is available here: <https://www.embopress.org/page/journal/14602075/authorguide#expandedview>.
6. Please upload the main and EV figures as individual production quality figure files in the .eps, .tif, or .jpg format (one file per figure).
7. Please compile the Appendix figures together with their legends into a single PDF file labelled "Appendix". Please preface the Appendix with a brief table of contents. The pages should be numbered.
8. CRedit has replaced the traditional author contributions section because it offers a systematic, machine-readable author contributions format that allows for more effective research assessment. Please remove the Authors Contributions from the manuscript and use the free text boxes beneath each contributing author's name in our online submission system to add specific details on the author's contribution. More information is available in our guide to authors.
9. Please rename "Conflict of interest" section into "Disclosure and competing interests statement" (further info: <https://www.embopress.org/page/journal/14602075/authorguide#conflictsofinterest>).
10. Please update references according to The EMBO Journal style - where there are more than 10 authors on a paper, the first 10 should be listed, followed by 'et al.' Please see further information here: <https://www.embopress.org/page/journal/14602075/authorguide#referencesformat>
11. Our data editors have flagged the following issues in figure legends that need correcting:
 - Please note that legend for figure 5g is missing or the figure legends of figure 5 are mislabeled. This needs to be rectified.
 - Please note that information related to n is missing in the legend of figures 1c; 4b, g, h, l; 5a; 6b, g, i, k, l
 - Please note that the error bars are not defined in the legend of figures 2c, f; 3e, g, i, j; 4b, d, e, g, h, j; 5a; 6b, c, e, g, i, k-m, p
 - Please note that scale bar and its definition are missing for figures 4a, i; 6h
 - Please note that the scale bar needs to be defined for figure 5b.
12. Papers published in The EMBO Journal are accompanied online by a 'Synopsis' to enhance discoverability of the manuscript. It consists of A) a short (1-2 sentences) summary of the findings and their significance, B) 3-4 bullet points highlighting key results and C) a synopsis image that is 550x300-600 pixels large (width x height, jpeg or png format). You can either show a model or key data in the synopsis image. Please note that the image size is rather small and that text needs to be readable at the final size. Please send us this information together with the revised manuscript.

Thank you again for giving us the chance to consider your manuscript for The EMBO Journal. I look forward to receiving the final version and your input on the source data issues.

With best wishes,

leva

leva Gailite, PhD
Senior Scientific Editor
The EMBO Journal
Meyerhofstrasse 1
D-69117 Heidelberg
Tel: +4962218891309
i.gailite@embojournal.org

We realize that it is difficult to revise to a specific deadline. In the interest of protecting the conceptual advance provided by the work, we recommend a revision within 3 months (9th Apr 2024). Please discuss the revision progress ahead of this time with the editor if you require more time to complete the revisions.

Referee #1:

The revised manuscript by Tanaka et al. has addressed my minor concerns with some new data that addresses potential mechanism (Fig 5) and strengthens the work, and rationale for conservative speculation. The work is comprehensive and rigorous, and it highlights a novel upstream transducer of fluid shear stress in endothelial cells. I have no further concerns and recommend publication.

Referee #2:

The authors have performed extensive revision and they have now addressed all major concerns. The reviewer supports the publication of this relevant work.

Referee #3:

The authors now show convincingly that there is a long term interaction between LPHN2 and VEC. Which already begins within minutes after the onset of flow. They further added in vitro experiments to show that LPHN2 is needed for flow-induced barrier stabilization. Both experiments are insightful and strengthen the manuscript.

On other points the response by the authors was not convincing.

- Most importantly: there is no detailed description of the mass spectrometry experiments, their results or statistic analysis. The authors mention that the raw data was uploaded in the data depository, but it is unclear which specific files the authors are referring to. Can they explain and demonstrate the results this more clearly?
- Methodology of the mass spec experiments are still not described in the methods section
- Results (beyond the gel) and quantifications of the mass spec detected peptides are still lacking from the manuscript.
- It is striking to read that the authors raise concerns about a paper in which the role of flow and Gaq/11 signaling in the aorta was previously demonstrated (Julián Albarrán-Juárez et al (JEM 2018)). The authors indicate that they think that the paper is "controversial", but do not explain what they mean by that or provide evidence of it. There is no reviewer of reader helped at all by this remark, and as such, that paper should be discussed appropriately within the context of the current work. In fact, the Albarran-Juarez paper already clearly showed that endothelial Gaq/11 is needed for cells to sense and align to flow. In the introduction, the authors do refer to Dela Paz et al for a notion about the disputed evidence for an upstream role of Piezo in flow induced Ga activation. Which is indeed relevant information, but also confusing since in the Dela Paz paper the respective authors show that flow rapidly disrupts the interaction between Gaq and Pecam-1, which goes seemingly against the findings in the manuscript by Tanaka, Schwartz et al. (Figure 3D). I do not understand why the authors chose not to discuss this in a transparent manner in the Discussion.
- Some suggestions made by multiple reviewers were not experimentally addressed as the authors mention that it is out of the scope. This refers to the retina assessment and appropriate permeability investigations in vivo. It is fully understandable that there is a limit to what is needed to substantiate the claims. However, in its current form some of the analysis in the mice and

zebrafish are preliminary. Comparing physiology with and without flow (silent heart, figure 4C, D, E) is interesting, but not physiological. The comments of the reviewers are aimed at strengthening the claims of the authors that "LPHN2 is required for flow-dependent remodeling in zebrafish and mice". If it is considered out of scope, then perhaps it is better to tone down the corresponding conclusions.

- The authors should consider moving the mention of SNPs associated with LPHN2 in the introduction or discussion section. In the current format (with conclusions in the abstract and results section) it seems as if the authors have performed experiments to demonstrate a link between the SNPs in LPHN2 and human disease.
- The GINIP_pulldown_LPHN2KD file deposited on Bioimage Archive contains a duplicated dataset (rows 7 to 11 are the same as rows 14 to 17) - please clarify
- 221106_cholesterol_measurement - There are multiple rows with the same numbers, it is not clear what these lanes represent as annotations are lacking. Please annotate shared raw data files appropriately. Sometimes there are only numbers provided, which makes it difficult for readers to interpret.

Please also clarify the issues with numerical duplications in the source data that are highlighted by referee #3. There are also further discrepancies I noted in the source data:

- Fig 1B: p-Src blot does not match the figure.

We apologize for the mistake. The file in the online database is now corrected.

- I cannot open source data files for Fig 1G-N, 2B, 2D-E, 3H, 3K, 4I, 5C - the downloaded files appear empty.

Sorry, not sure what caused the problem but these files are now reuploaded.

- Unexplained duplicate values have been found also in source data for figures 3L (please note column G), 5A, 6G (summary tab) and 6OP. I have attached the corresponding files with the detected duplications labelled in red. Please take a look and correct as needed. A brief explanation would be very helpful.

We corrected the source data of Figure 3L, Figure 5A. In Figure 6G, the summary tab shows the left top is control thigh, the left bottom is Lphn2ECKO thigh. The top right is control calf and bottom right is Lphn2 ECKO calf. Since it is confusing, we organized the raw data to visualize clearly.

Reply to reviewer 3:

The authors now show convincingly that there is a long term interaction between LPHN2 and VEC. Which already begins within minutes after the onset of flow. They further added in vitro experiments to show that LPHN2 is needed for flow-induced barrier stabilization. Both experiments are insightful and strengthen the manuscript.

We thank the reviewer for this positive evaluation.

On other points the response by the authors was not convincing.

- Most importantly: there is no detailed description of the mass spectrometry experiments, their results or statistic analysis. The authors mention that the raw data was uploaded in the data depository, but it is unclear which specific files the authors are referring to. Can they explain and demonstrate the results this more clearly?

To address this issue, we show the mass spec results in the protein list in the Excel files provided by the Yale core facility. Please note that detailed peptide quantification was not done. As shown in Appendix Figure 4E, the provided protein list was derived from the extracted band. Each file name corresponds to the molecular weight of the protein.

- Methodology of the mass spec experiments are still not described in the methods section

We now describe in detail how samples were prepared for the mass spec experiments in the Methods section of the manuscript.

- Results (beyond the gel) and quantifications of the mass spec detected peptides are still lacking from the manuscript.

As written above, we received the protein list from Yale core facility which did not include quantification of mass spec detected peptides. We confirmed that emPAI (exponentially modified Protein Abundant Index) for ADGRL3 and S1PR1 are 0.02 and 0.07, respectively. To support mass spec findings, we performed coimmunoprecipitations using the same Ga construct under the same flow conditions, which confirmed the interaction between Latrophilin and the Ga variant in flow-dependent manner.

- It is striking to read that the authors raise concerns about a paper in which the role of flow and Gaq/11 signaling in the aorta was previously demonstrated (Julián Albarrán-Juárez et al (JEM 2018)). The authors indicate that they think that the paper is "controversial", but do not explain what they mean by that or provide evidence of it. There is no reviewer of reader helped at all by this remark, and as such, that paper should be discussed appropriately within the context of the current work. In fact, the Albarran-Juarez paper already clearly showed that endothelial Gaq/11 is needed for cells to sense and align to flow. In the introduction, the authors do refer to Dela Paz et al for a notion about the disputed evidence for an upstream role of Piezo in flow induced Ga activation. Which is indeed relevant information, but also confusing since in the Dela Paz paper the respective authors show that flow rapidly disrupts the

interaction between Gαq and Pecam-1, which goes seemingly against the findings in the manuscript by Tanaka, Schwartz et al. (Figure 3D). I do not understand why the authors chose not to discuss this in a transparent manner in the Discussion.

This was a minor concern in the first review. We have now expanded the Discussion to talk about other G protein mechanisms at greater length, including the Dela Paz results. But we hope that the reviewer appreciates that Piezo1 is not a component of our paper in any way, thus, a detailed discussion of the issues and controversies around Piezo1 does not belong.

Regarding the controversies related to Albarrán-Juárez et al, we apologize for not being clearer. This comment was colored by additional unpublished data from David Beech and my lab that raise questions about findings from Albarran-Juarez. We are now completing the last experiments for a joint paper and cannot present that work here. This review article (The Interplay of Endothelial P2Y Receptors in Cardiovascular Health: From Vascular Physiology to Pathology - PubMed (nih.gov)) does a good job of discussing the many effects and controversies surrounding P2Y2 in the vascular endothelium. The reviewer may wish to note that P2Y2 signals via both G protein dependent and independent mechanisms. In any case, there are complicated and uncertain issues related to Albarrán-Juárez et al. As they are not directly relevant to our findings, we don't feel that we can address them in a productive way until our Piezo1 experiments are completed.

- Some suggestions made by multiple reviewers were not experimentally addressed as the authors mention that it is out of the scope. This refers to the retina assessment and appropriate permeability investigations in vivo. It is fully understandable that there is a limit to what is needed to substantiate the claims. However, in its current form some of the analysis in the mice and zebrafish are preliminary. Comparing physiology with and without flow (silent heart, figure 4C, D, E) is interesting, but not physiological. The comments of the reviewers are aimed at strengthening the claims of the authors that "LPHN2 is required for flow-dependent remodeling in zebrafish and mice". If it is considered out of scope, then perhaps it is better to tone down the corresponding conclusions.

We have discussed this issue with the editor who agrees that further experimental work examining tissues and vessels is not required. We are well aware that stopping the heart in zebrafish embryos is not physiological but it is a widely used and reliable experimental approach for determining whether and event is dependent on biomechanical forces from blood flow.

- The authors should consider moving the mention of SNPs associated with LPHN2 in the introduction or discussion section. In the current format (with conclusions in the abstract and results section) it seems as if the authors have performed experiments to demonstrate a link between the SNPs in LPHN2 and human disease.

The sentence in the abstract is "Human genetic data reveal a correlation between the *Adgrl2* gene and cardiovascular disease.". We have discussed this with the editor who agrees that readers are unlikely to construe this to mean that we have done experiments to demonstrate a functional link. Further, the Discussion makes it clear that we haven't.

- The GINIP_pulldown_LPHN2KD file deposited on Bioimage Archive contains a duplicated dataset (rows 7 to 11 are the same as rows 14 to 17) - please clarify
- 221106_cholesterol_measurement -There are multiple rows with the same numbers, it is not clear what these lanes represent as annotations are lacking. Please annotate shared raw data files appropriately. Sometimes there are only numbers provided, which makes it difficult for readers to interpret.

We apologize for the confusion. We have updated the raw data file appropriately and uploaded to data repository

Dear Martin,

Thank you for addressing the remaining source data issues. I am now pleased to inform you that your manuscript has been accepted for publication.

Before we forward your manuscript to our publishers, there are a couple of points that I would need your input on:

- 1) Upon resizing to the required dimensions (550 pixels width x 350-600 pixels height), the synopsis image becomes largely unreadable. Could you please adjust the image, keeping the final dimension requirements in mind?
- 2) I would like to propose some minor edits in the manuscript title, abstract and synopsis (please see below and the attached manuscript text file). I have also written a short blurb that will accompany the title of your manuscript in our online system. Please let me know if any corrections are needed:

Title:

Latrophilin-2 mediates fluid shear stress mechanotransduction at endothelial junctions

Blurb:

Shear stress-induced activation of G α i2 and G α q/11 by the adhesion receptor latrophilin-2 ensures PECAM-1 signaling and vascular remodeling in zebrafish and mice.

Synopsis

Fluid shear stress from blood flow induces vascular endothelium remodeling by triggering signaling events through the PECAM-1/VE-cadherin/VEGF receptor complex at cell-cell junctions. This study shows that the adhesion G protein-coupled receptor latrophilin-2 (Lphn2) activates G α i2 and G α q/11 to induce shear stress-dependent activation of this pathway.

- Shear stress-induced activation of G α i2 and G α q/11 is required for endothelial cell alignment and activation of junctional signaling events.
- Lphn2 is the upstream GPCR that co-localizes and interacts with PECAM-1 to activate its signaling.
- Deletion of Lphn2 from zebrafish and mice blocks multiple flow-dependent developmental and adult vascular remodeling processes.

If you have any questions, please do not hesitate to contact the Editorial Office. Thank you for this contribution to The EMBO Journal and congratulations on a successful publication!

With best wishes,

leva

leva Gailite, PhD
Senior Scientific Editor
The EMBO Journal
Meyerhofstrasse 1
D-69117 Heidelberg
Tel: +4962218891309
i.gailite@embojournal.org
